# Molecular characterization of the sea lamprey retina illuminates the evolutionary origin of retinal cell types

Junqiang Wang ⬡, Lin Zhang ⬡, Martina Cavallini ⬡, Ali Pahlevan, Junwei Sun, Ala Morshedian, Gordon L. Fain ⬡, Alapakkam P. Sampath ⬡ & Yi-Rong Peng ⬡ ✉

The lamprey, a primitive jawless vertebrate whose ancestors diverged from all other vertebrates over 500 million years ago, offers a unique window into the ancient formation of the retina. Using single-cell RNA-sequencing, we characterize retinal cell types in the lamprey and compare them to those in mouse, chicken, and zebrafish. We find six cell classes and 74 distinct cell types, many shared with other vertebrate species. The conservation of cell types indicates their emergence early in vertebrate evolution, highlighting primordial designs of retinal circuits for the rod pathway, ON-OFF discrimination, and direction selectivity. The diversification of amacrine and some ganglion cell types appears, however, to be distinct in the lamprey. We further infer genetic regulators in specifying retinal cell classes and identify ancestral regulatory elements across species, noting decreased conservation in specifying amacrine cells. Altogether, our characterization of the lamprey retina illuminates the evolutionary origin of visual processing in the retina.

The complex structure of the vertebrate nervous system arises from a distinct array of cell types present within each processing center. Much effort has recently been given to classifying cell types and describing their molecular differences[1,2]. Our understanding of neuronal cell-type specification is nowhere more advanced than in the vertebrate retina, where the description of its layered structure containing photoreceptors, interneurons, and ganglion cells dates back to the pioneering work of Ramón y Cajal[3]. More recent experimentation has identified over 100 cell types in mammals distributed into six cell classes, and much is known about the anatomy, development, and physiology of these different cell types[4,5]. It is striking that the fundamental structure of the retina is conserved across all vertebrate lineages[6,7], which seems to indicate an ancient origin; but the cellular and molecular blueprint for the evolutionary formation of the retina remains largely unknown.

We hoped to shed some light on the evolution of the retina by studying the lamprey, a jawless vertebrate (agnathan) whose progenitors diverged from all other vertebrates in the Cambrian over 500 million years ago (Fig. 1a)[8,9]. The eye of the adult lamprey shows remarkable similarities in structure to the conserved camera eye of jawed vertebrates (gnathostomes), with a cornea, lens, pigmented epithelium, and retina with laminar structure[10,11]. Moreover, the lamprey retina shares key anatomical features with other vertebrates, containing three cellular layers interconnected within two synaptic plexiform layers (Fig. 1b)[11,12]; and it is duplex, with functionally distinct rods and cones[13,14]. There are, however, some remarkable differences, which may suggest a primitive state for the lamprey retina. First, the rods in the lamprey are morphologically similar to cones, with an outer segment consisting of invaginating lamellae continuous with the plasma membrane[10,15]. Second, retinal ganglion cells are located on both sides of the inner plexiform layer, with their axons forming the optic fiber layer between the inner plexiform layer and the inner nuclear layer, instead of at the vitread margin of the retina as in other vertebrates (Fig. 1b). Finally, the three-layered retina is acquired only after the lamprey undergoes metamorphosis[12,16,17].

High throughput single-cell RNA-sequencing (scRNA-seq) has become a pivotal tool for characterizing cell types in the nervous system from various vertebrate and invertebrate species[18–21]. This approach

Department of Ophthalmology and Stein Eye Institute, UCLA David Geffen School of Medicine, Los Angeles, CA, USA.
✉ e-mail: yirongpeng@mednet.ucla.edu

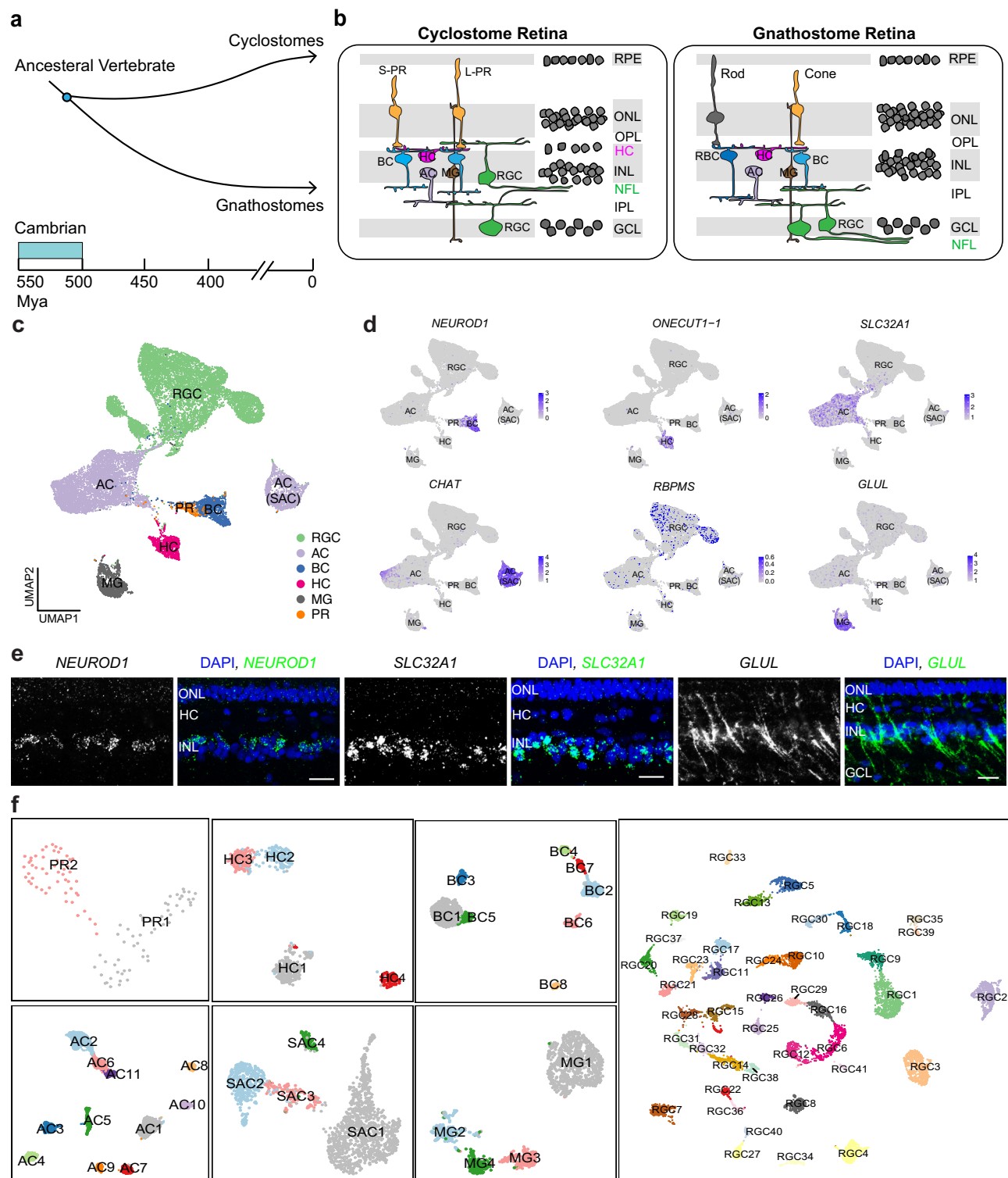

**Fig. 1 | Cell atlas of the adult lamprey retina. a** Evolutionary history of vertebrates illustrating the split between cyclostomes (jawless) and all other vertebrates (gnathostomes) during the Cambrian period over 500 million years ago (MYA). **b** Sketches of cellular arrangement and retinal circuitry in gnathostome (right) and cyclostome, particularly lamprey (left). Major differences include the position of NFL (highlighted in green), the distinct somatic layer of HCs, and the displaced RGCs between the two groups. **c** Uniform Manifold Approximation and Projection (UMAP) visualization of six lamprey cell classes, derived from a dataset of 21,474 cells. **d** Feature plots displaying the expression of canonical markers within individual cell classes. **e** Fluorescence in situ hybridization (FISH) validation of

*NEUROD1* (green) in bipolar cells, *SLC32A1* (green) in amacrine cells, and *GLUL* (green) in Müller glia. Nuclei stained with DAPI are in blue. Scale bar, 20 μm. Each experiment was performed independently three times with similar results. **f** UMAP visualizations of various cell types within each cell class, with SACs separated from the other ACs. Abbreviations: AC, amacrine cell; SAC, starburst amacrine cell; BC, bipolar cell; HC, horizontal cell; MG, Müller glia; PR, photoreceptor; S-PR, short photoreceptor; L-PR, long photoreceptor; RGC, retinal ganglion cell; GCL, ganglion cell layer; IPL, inner plexiform layer; INL, inner nuclear layer; NFL, nerve fiber layer; ONL, outer nuclear layer; OPL, outer plexiform layer.

offers a comprehensive assessment of genomic activity within individual cell types. By comparing gene-expression patterns of cell types across species, we can construct homologous relationships and trace the evolutionary trajectory of cell-type differentiation[22,23]. Notably, a recent study focusing on 13 mammalian species revealed a striking homologous relationship among cell types within bipolar cell and ganglion cell classes, suggesting an ancestral origin for cell-type diversification among mammals[21]. Extending this approach to the lamprey could provide a much deeper understanding of the evolutionary origin of cell types in the vertebrate retina. The comparison of cell types across a large phylogenetic distance nevertheless presents many challenges, resulting from incomplete genome and gene annotation of non-model organisms, distinct adaptations of model species to their environments, and the loss of homologous genes during species divergence[21,24,25]. Moreover, cell types are specified by transcriptional regulatory programs, which control genomic accessibility in each cell type[26]. Evaluating the conservation of these regulatory mechanisms between lamprey and jawed vertebrates can shed light on the genetic underpinnings of cell-type evolution in the vertebrate retina.

In this study, we used scRNA-seq to characterize the cell types in the adult retina of the sea lamprey (*Petromyzon marinus*). To facilitate cross-species comparison, we constructed a retina-specific transcriptome, which enhances the annotation of coding genes specific to lamprey retina tissue. Our scRNA-seq data revealed a total of 74 distinct cell types from all six retinal cell classes. Comparative analyses of lamprey cell types against those from other vertebrate species revealed multiple conserved cell types, including, among others, rod and cone photoreceptors, rod bipolar cells, AII-like amacrine cells, Type I and Type II horizontal cells, ON and OFF starburst amacrine cells, and direction-selective ganglion cells. These results indicate that the foundational circuitry for specific features of light detection and signal integration in the retina emerged among the very earliest vertebrates, likely before the split of progenitors of lamprey and other cyclostomes from the vertebrate lineage. To investigate shared genetic regulatory elements across phylogenetic distance, we developed a network-based methodology to infer the activities of specific proteins, including transcription factors (TFs), transcription cofactors (coTFs), and surface and signaling proteins, in individual cell classes. By comparing these genetic regulators across species, we not only identified genetic programs likely inherent to the whole vertebrate lineage but also illustrated the distinct conservation of their utilization patterns among cell classes. Our work has provided insights into the evolution of retinal cell types and the mechanisms by which type specifications were first established, forming the basis of light detection in the vertebrate eye.

## Results

### Cell atlas of the adult sea lamprey retina
To facilitate cell and gene discovery, we used TruSeq to generate a retina-specific transcriptome of the lamprey (see "Methods"). We first compared the lamprey genome references from Ensembl (Pmarinus_7.0) and NCBI (kPetMar1.pri)[27]. Of the two, kPetMar1 yielded higher mapping percentages of reads to genome, exons, and transcriptome when used for alignment (Supplementary Fig. 1a). We thus built a retina-specific transcriptome based on the kPetMar1.pri version, named as "NCBI+TruSeq." Using NCBI+TruSeq transcriptome files, we further improved mapping metrics (Supplementary Fig. 1a). The NCBI +TruSeq reference transcriptome has 15,070 genes updated by StringTie with MSTRG numbers, suggesting changes in their gene body definitions (Supplementary Fig. 1c). Notably, many of these updated genes are crucial for the physiological function of retinal cells. For example, the cone *red-opsin* gene was updated with an additional 3' exon, where abundant reads from both TruSeq and single-cell RNA-seq were aligned (Supplementary Fig. 1b). Moreover, a detailed comparison between the NCBI and NCBI+TruSeq references using gffcompare

revealed 15,871 novel exons, 11,391 novel introns, and 3430 novel loci in the NCBI+TruSeq reference transcriptome. Although the NCBI assembly kPetMar1.pri is a chromosome-level genome assembly, over 60% of annotated genes were named with a "LOC" number without a gene symbol. We further annotated these LOC genes to correlated gene symbols based on the information of their gene products (see "Methods", Supplementary Data 1). The final reference transcriptome has nearly 70% of its genes with specific gene symbols, thus greatly improving our ability to classify cell types and characterize molecular profiles of lamprey retinal cells (Supplementary Data 1).

After mapping scRNA-seq reads to retina-specific transcriptomes, we obtained 21,474 high-quality transcriptomes from individual cells, with 12,021 and 9,453 cells in each respective replicate. Using an unsupervised clustering method, we identified six main cell classes: photoreceptors (PR), horizontal cells (HC), bipolar cells (BC), amacrine cells (AC), retinal ganglion cells (RGC), and Müller glia (MG) (Fig. 1c)[28]. We did not detect other, non-neuronal cell types from our isolated-retina preparations (see "Methods"), such as retinal pigment epithelium, pericytes, and microglia (Supplementary Fig. 1d). No bias associated with biological replicates was observed (Supplementary Fig. 2a, b). Each lamprey cell class showed the expression of a similar set of marker genes, which are also expressed in the mouse or macaque retina (Fig. 1d)[29,30]. The expression of these canonical cell-class markers was confirmed by fluorescence in situ hybridization in the lamprey retina (Fig. 1e). We further clustered cell types from individual cell classes, with ACs separating into starburst amacrine cells (SACs) and other AC subclasses (Fig. 1f). We identified a total of 2 PR, 4 HC, 8 BC, 4 SAC AC, 11 non-SAC AC, 41 RGC, and 4 MG clusters (Fig. 1f). All cell types were present in each replicate (Supplementary Fig. 2c). Using hierarchical clustering, we found that each cell type from the same cell class was more closely correlated to cell types within their class than to types outside of their class (Supplementary Fig. 2d, see "Methods"). Thus, the lamprey retina is composed of at least 74 cell types with clear molecular distinction.

### Conserved photoreceptor cell types between the lamprey and jawed species
The number of diverse cell types identified in the lamprey is comparable to that seen in jawed vertebrate species[4,5], suggesting that cell-type diversification might have originated with the very earliest of vertebrates. To explore this hypothesis in greater detail, we compared lamprey cell types to those in three jawed vertebrate species–mouse, chicken, and zebrafish, whose retinal cell types have in most cases been well characterized from scRNA-seq (Supplementary Table 1)[19,31–35]. We employed two approaches to achieve this comparison. 1) We integrated lamprey cell types in individual classes with cell types in selected species. 2) We used transcriptomic mapping via XGBoost[36] to identify the closest related cell types in selected jawed species. We found strikingly conserved cell types, including photoreceptors, bipolar cells, horizontal cells, and starburst amacrine cells, shared between lamprey and other species, as well as interesting divergences. We present these findings in the following.

We identified two lamprey photoreceptors (PR): PR1 and PR2 (Fig. 2a). The expression of *rhodopsin* in PR1 and cone *red-opsin* in PR2 identify these morphologically defined "short" and "long" PRs as a rod and a single type of cone (Fig. 2b, c and Supplementary Fig. 3), in agreement with results from microspectrophotometry, physiological recordings and molecular validations[13,14,37–41]. The functional differences between rods and cones are achieved through the use of distinct molecules in their phototransduction cascades[42,43]. Interestingly, we found strikingly similar differences in the phototransduction cascades between lamprey PR1 and PR2 (Fig. 2b)[11]. We further confirmed that the G-protein subunit alpha transducin 1 (*GNAT1*) is expressed in PR1, while *GNAT2* is expressed in PR2 (Fig. 2c, Supplementary Table 2)[44].

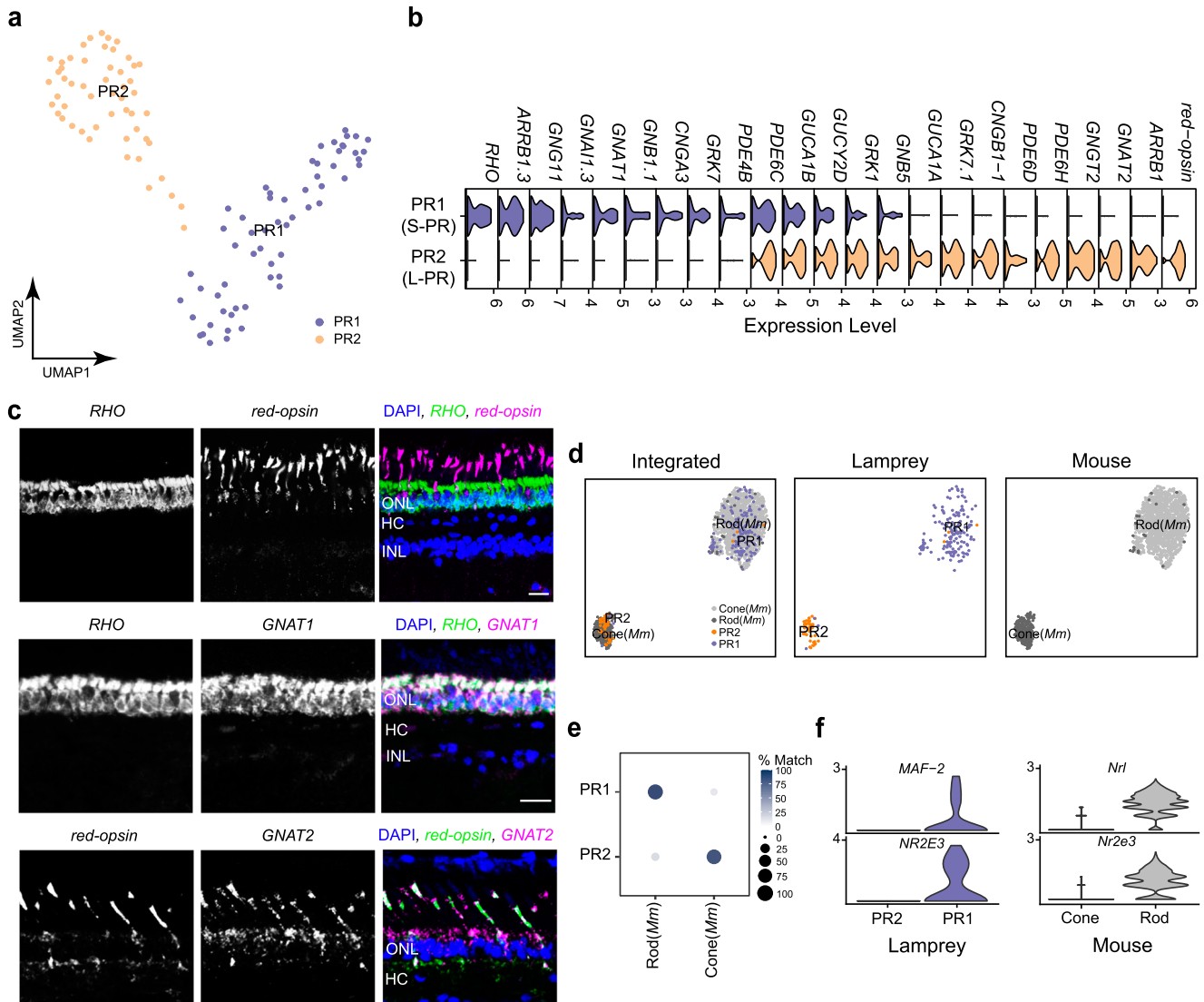

**Fig. 2 | Classification of lamprey PR types and their comparison with mouse PR types. a** UMAP visualization of two lamprey PR types. **b** Stacked violin plot showing distinct gene-expression patterns in the phototransduction cascade between PR1 (S-PR) and PR2 (L-PR). S-PR, short-photoreceptor (rod); L-PR, long-photoreceptor (cone). **c** FISH validations showing exclusive expression of *rho-dopsin* (*RHO*, green in the merged image) and *red-opsin* (magenta in the merged image) between PR1 and PR2 (upper panel), co-expression of *RHO* (green in the merged image) and *GNAT1* (magenta in the merged image) in PR1 (middle panel), and co-expression of *red-opsin* (green in the merged image) and *GNAT2* (magenta in the merged image) in PR2 (bottom panel). Nuclei stained with DAPI are in blue.

Scale bar, 20 μm. Each experiment was performed independently three times with similar results. **d** Integration of lamprey and mouse PRs visualized with UMAP, with both integrated and species-specific clusters presented in separated UMAP plots. **e** Confusion matrix demonstrating the transcriptomic correspondence of PR types between lamprey and mouse (*Mus musculus, Mm*). Mouse PRs were used as the training dataset, while lamprey PRs were used as the testing dataset. Circles and color gradients present the percentage of cells from a lamprey PR cluster assigned to a corresponding mouse PR type. See Source Data. **f** Violin plots showing the expression of conserved transcription factors enriched in both lamprey PR1 and mouse rods.

To assess the molecular similarities between lamprey and mouse photoreceptors, we compared their transcriptomes. We integrated mouse PRs[31] with lamprey PRs and identified two clusters through an unsupervised method (Fig. 2d, Supplementary Table 1). As expected, lamprey PR1 aligned well with mouse rods, while PR2 aligned with mouse cones (Fig. 2d). These findings are supported by transcriptomic mapping via XGBoost, which shows that PR1 corresponds to mouse rods and PR2 to mouse cones (Fig. 2e).

Different transcription factors distinguish rod from cone fate in the mouse retina, such as *Nrl* and *Nr2e3* (Fig. 2f)[45–48]. Notably, we found that either the same gene (*NR2E3*) or a gene from the same NRL-MAF-family (*MAF-2*) is differentially expressed in PR1 and PR2[49]. These findings, taken together, show that PR1 and PR2 are molecularly similar to rods and cones, and that their differentiation is

likely regulated by at least partially conserved transcriptional mechanisms.

### Conserved bipolar cell types between the lamprey and jawed species

We identified eight BC types (Fig. 3a). BCs are typically categorized into ON and OFF subclasses based on expression of metabotropic and ionotropic glutamate receptors[50]. By studying the genes encoding these channels, we found that lamprey BCs can be similarly classified into ON types expressing the group III metabotropic glutamate receptor *GRM8.2,* and OFF types expressing the kainate receptor *GRIK2-2* or *GRIK2* (Fig. 3b, Supplementary Fig. 4a). Using OrthoFinder to compare protein sequences between lamprey and five jawed species (zebrafish, chicken, mouse, macaque, and human), we found that the

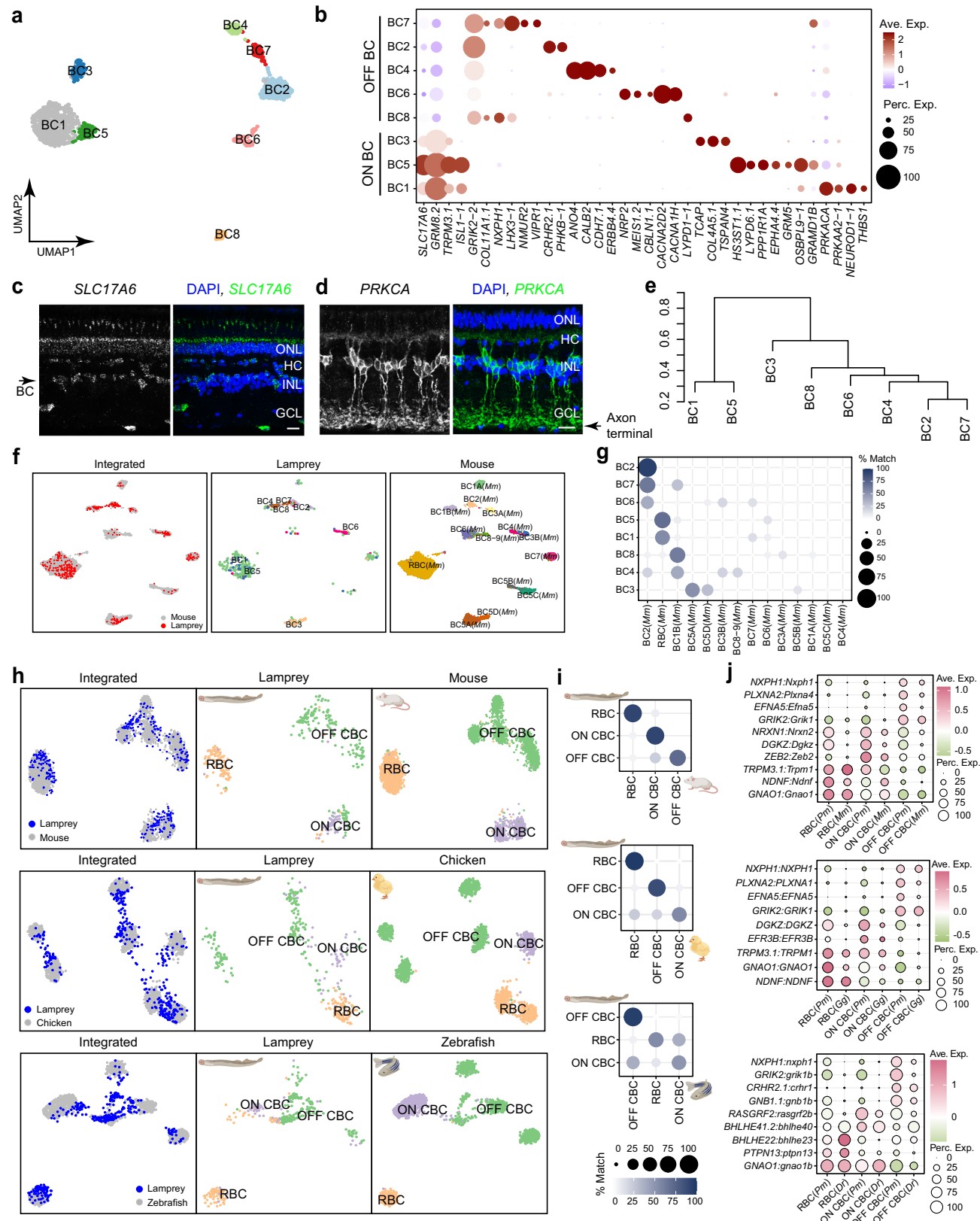

lamprey *GRIK2* gene is an ortholog of *GRIK1* in jawed species, which is typically expressed in OFF BCs in these jawed species (Supplementary Fig. 4c, d)[33,51]. Moreover, lamprey ON BC types expressed *TRPM3.1*—a type of transient receptor potential cation channel, which could couple with *GRM8.2* in the lamprey (Fig. 3b, Supplementary Fig. 4a).

Although *TRPM1* and *GRM6*, a classically coupled channel pair in mouse ON BCs, are not found in the lamprey genome, lamprey *TRPM3.1* and *GRM8.2* appear to be orthologous to mouse *TRPM1* and *GRM6* (Supplementary Fig. 4e, f). Thus, lamprey BC types can be molecularly classified into ON and OFF subclasses, distinguished by

**Fig. 3 | Classification of lamprey BC types and their comparison with jawed BC types. a** UMAP visualization of eight BC clusters in the lamprey retina. **b** Dot plot showing expression patterns of markers specific to ON and OFF BC subgroups, as well as markers unique to individual BC types. Perc. Exp., percentage of expression; Ave. Exp., average expression. See source data. **c** FISH validation of *SLC17A6* (green) expression in multiple cell types in the lamprey, including BC (arrow). Nuclei stained with DAPI are in blue. Scale bar, 20 μm. Experiments were performed independently three times with similar results. **d** Immunostaining with anti-PRKCA antibody in the lamprey retina showing the morphology of protein kinase C expressing rod bipolar cells (RBCs), with axon terminals of RBCs indicated by an arrow. Scale bar, 20 μm. Experiments were performed independently three times with similar results. **e** Complete linkage agglomerative hierarchical clustering of BC types from correlation distance. The scale on the left indicates correlation distance.

**f** Integration of lamprey and mouse BCs visualized with UMAP, with both integrated and species-specific clusters presented in separated UMAP plots. **g** Confusion matrix demonstrating the transcriptomic correspondence of BC types between lamprey and mouse. Mouse (*Mus musculus, Mm*) BCs were used as the training dataset, while lamprey BCs were used as the testing dataset. See Source Data. **h** UMAP visualizations of the integrated BC subclasses (RBC, ON CBC, and OFF CBC) between lamprey and each jawed species (mouse, chicken, and zebrafish), with integrated and species-specific clusters in separated plots. *Dr, Danio rerio. Pm, Petromyzon marinus*. **i** Confusion matrix demonstrating the transcriptomic correspondence of BC subclasses between lamprey and each jawed species. See Source Data. **j** Dot plots showing expression patterns of pairs of orthologous genes across conserved BC subclasses. See Source Data. Ave. Exp., average expression; Perc. Exp., percentage of expression.

expressions of similar glutamate receptors. However, unlike mammalian BCs, lamprey ON BCs expressed *SLC17A6*, a glutamate transporter in mammals specific to retinal ganglion cells (Fig. 3b, c)[29,30,34,52,53].

Given that rods and cones are molecularly distinct in the lamprey retina, we explored the possibility of classifying BC types into rod BCs (RBCs) and cone BC types (CBCs)[33,54]. In mammals, protein kinase C alpha (*PrkCa*) and *Gramd1b* are markers for RBCs[30,33]. We found that *PRKACA* (an alias of *PRKCA*)[55] is highly expressed in BC1, and *GRAMD1B* is highly expressed in BC5 and BC7. From the glutamate receptors expressed by these cells (Fig. 3b), we classified BC1 and BC5 as ON and BC7 as OFF. From hierarchical clustering and correlation-expression analysis, we observed that BC1 and BC5 are closely related and stand apart from the other types (Fig. 3e, Supplementary Fig. 4b). We suspect that both of these cells are ON RBCs. Unlike mammals, lamprey have a rod bipolar cell that is hyperpolarizing, which could be BC7[56]. Using anti-PRKCA antibody to stain the lamprey retina, we found that PRKCA-positive BCs resemble mouse RBCs morphologically. The cell bodies of these BCs press against the outer plexiform layer, while their axons innervate the innermost part of the inner plexiform layer (Fig. 3d). It is likely that these are the BC1 cells.

We next compared the transcriptomic profiles of lamprey BC types with those in mice, as their RBCs have been extensively studied[57–60]. We integrated mouse BCs with lamprey BCs with either canonical correlation analysis (CCA) or reciprocal PCA (RPCA)[61]. Interestingly, lamprey BC1 and BC5 integrate well with mouse RBCs in both methods (Fig. 3f, Supplementary Fig. 4g, h). The other ON BC, BC3, aligns with mouse ON cone bipolar cells (ON CBCs) BC5A and BC5D; while all lamprey OFF BC types integrate with mouse OFF cone bipolar cells (OFF CBCs) BC1B or BC2 (Fig. 3f). We further used transcriptomic mapping via XGBoost to correlate lamprey BC types to the 15 mouse BC types. BC1 and BC5 were consistently analogous to mouse RBCs, with the remainder of lamprey bipolar cells corresponding either to ON CBCs (BC5A or BC5D) or to OFF CBCs (BC1B or BC2) (Fig. 3g, Supplementary Fig. 4h).

Our discovery of RBCs in the lamprey indicates a Cambrian origin for this cell type. To confirm this finding, we expanded the comparison from mice to zebrafish and chicken[19,35]. We first integrated zebrafish and chicken BC types with mouse ones to verify the robustness of our cross-species comparison methods, considering the different levels of gene duplications in individual species (Supplementary Fig. 4c)[62]. Consistent with a reported study[35], zebrafish RBC1 (c14) and RBC2 (c19) are transcriptomically related to mouse RBCs, with RBC1 showing the highest correspondence (Supplementary Fig. 5a–c). We also identified RBCs, ON CBCs, and OFF CBCs in chicken (Supplementary Fig. 5d–f). With the establishment of the three subclasses (RBCs, ON CBCs, and OFF CBCs) in each jawed species, we then integrated them with lamprey BCs. We found that RBCs, ON CBCs, and OFF CBCs from each jawed species consistently align with the corresponding lamprey subclasses, which is further supported by transcriptomic mapping results (Fig. 3h, i). Furthermore, we identified orthologs that serve as shared markers in each subclass between lamprey and each jawed

species, with some of these marker genes being common across vertebrates. For example, *NXPH1* is a common OFF CBC gene across the four species (Fig. 3j). Despite the conservation, each species also acquired specific genes in these shared subclasses (Supplementary Fig. 5g–i). Thus, our results underscore that, akin to mammals and other jawed vertebrates, lamprey BC types are primarily divided into RBCs (which in lamprey can be either ON or OFF), ON CBCs, and OFF CBCs. It is particularly noteworthy that lamprey RBCs might comprise as many as three types[56].

## Conserved horizontal cell types between the lamprey and jawed species

We identified four horizontal cell (HC) types in the lamprey retina (Fig. 4a). Nearly all sea lamprey HCs express the melanopsin-like gene *OPN4l.1* (Fig. 4b, c). The expression pattern is similar to that previously reported in the river lamprey (*Lethenteron camtschaticum*)[63]. Like chicken and macaque Type I and Type II HCs, lamprey HCs can be divided into two subclasses, each exclusively expressing either *ISL1* or *LHX1* (Fig. 4c)[19,30]. We further integrated data from lamprey HCs with chicken HCs[19]. In the chicken retina, there are five HC types: HC1/3 are classified as Type I HCs, while HC2/4/5 belong to Type II HCs. Our integrated data separate into two clusters: lamprey H1 and H4 align with chicken HC2/4/5, and lamprey H2 and H3 align with chicken H1/3 (Fig. 4d). Using transcriptomic mapping via XGBoost, we obtained a corresponding result mirroring the integrated pattern (Fig. 4e). Moreover, we identified conserved marker genes between lamprey HC1/4 and chicken HC2/4/5, as well as between lamprey HC2/3 and chicken HC1/3 (Fig. 4f). We conclude that the four HC types in the lamprey correspond to the Type I and Type II HCs found in chicken.

## Conserved starburst amacrine cell types between the lamprey and jawed species

Starburst amacrine cells (SACs) are cholinergic cells in the retina, which are essential components of retinal circuits detecting the direction of motion[64–66]. SACs constitute ~22% of all the amacrine cells in the lamprey retina, a proportion much greater than the 5% in the mouse retina[67]. We identified four SAC clusters (Fig. 4g), all of which express canonical SAC markers, such as *SOX2*, *CHAT*, and *MEGF10* (Fig. 4h, i)[68,69]. Interestingly, SAC1 and SAC3 were found to express *FEZF2* (Fig. 4i), a gene from the same family as *Fezf1*, which in the mouse retina determines the fate of ON SACs[32]. These findings suggest that SAC types in lamprey might also be classified into ON and OFF subgroups. In fact, when integrating data from lamprey SACs with mouse ones, we observed that lamprey SAC1 and SAC4 aligned more closely with mouse ON SACs, while SAC2 and SAC3 were more closely aligned with mouse OFF SACs (Fig. 4j). Through transcriptomic mapping via XGBoost, we determined that among all the SAC types, SAC1 and SAC2 most closely correspond to mouse ON SACs and OFF SACs, respectively (Fig. 4k). Moreover, they shared multiple conserved markers with mouse ON or OFF SACs (Fig. 4l). Therefore, our results indicate that lamprey SACs include both ON and OFF types, and that

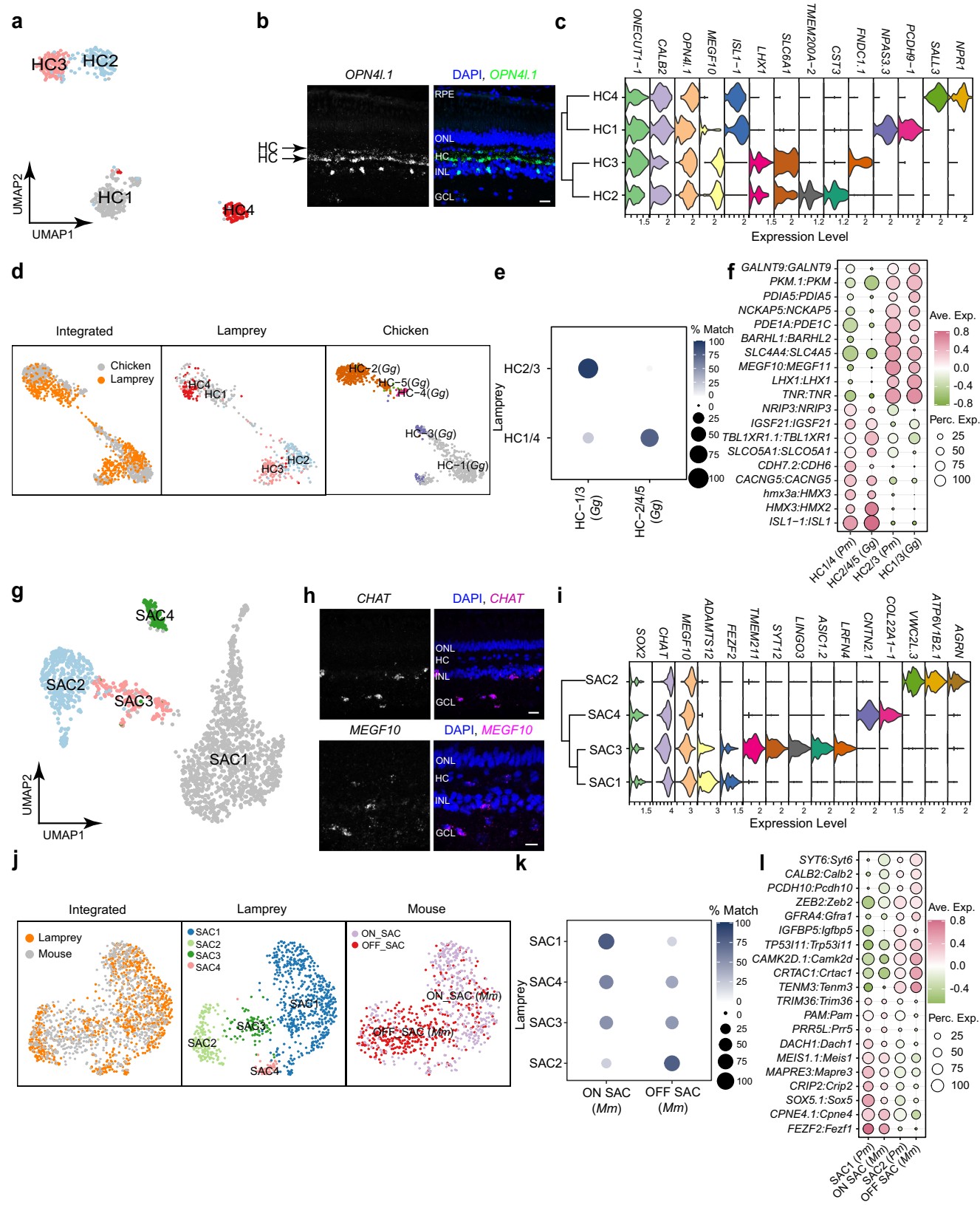

their fates may be determined at least in part by members of the *FEZ* gene family.

## The evolutionary diversification of AC types

The remarkable conservation across PR, BC, HC, and SAC types in the lamprey hints at a foundational program that is at the core of the structure of all vertebrate retinas. Our exploration now broadens to include the other two cell classes–ACs and RGCs, which across jawed vertebrates display the most heterogeneity among cell classes[19,21,30,34]. We found that the lamprey AC and RGC classes also contain the most heterogeneous cell types compared to other classes. We related AC and RGC types from the lamprey to those in mouse, chicken, and

**Fig. 4 | Classification of lamprey HC and SAC types and their comparison with chicken or mouse types. a** UMAP visualization of four lamprey HC types. **b** FISH validation of the expression of the melanopsin-like gene, *OPN4l.1* (green in the merged image), in the lamprey retina. Arrows point to HC layers. Nuclei stained with DAPI are in blue. Scale bar, 20 μm. Experiments were performed independently three times with similar results. **c** Stacked violin plot showing expression patterns of common markers for all HC types and markers specific to individual HC types. The dendrogram on the left shows agglomerative hierarchical clustering of HC types. **d** Integration of lamprey and chicken HCs visualized with UMAP, with both integrated and species-specific clusters presented in separated UMAP plots. **e** Confusion matrix demonstrating transcriptomic correspondence of HC types between lamprey and chicken (*Gallus gallus*, *Gg*). Chicken HCs were used as the training dataset, while lamprey HCs were used as the testing dataset. See Source Data. **f** Dot plot showing expression patterns of conserved orthologous genes across corresponding HC types in lamprey and chicken (*Gg*). See Source Data.

**g** UMAP visualization of four lamprey SAC types. **h** FISH validation of the expression of *CHAT* (upper panel, magenta in the merged image) and *MEGF10* (bottom panel, magenta in the merged image) in the lamprey retina. Nuclei stained with DAPI are in blue. Scale bar, 20 μm. Each experiment was performed independently three times with similar results. **i** Stacked violin plot showing markers common to all SAC types and markers specific to individual SAC types. The dendrogram on the left shows agglomerative hierarchical clustering of SAC types. **j** Integration of lamprey and mouse SACs visualized with UMAP, with both integrated and species-specific clusters presented in separated UMAP plots. **k** Confusion matrix demonstrating transcriptomic correspondence of SAC types between lamprey and mouse. Mouse (*Mm*) SACs were used as the training dataset, while lamprey SACs were used as the testing dataset. See Source Data. **l** Dot plot showing the expression patterns of conserved orthologous genes across corresponding SAC types in lamprey and mouse (*Mm*). See Source Data.

zebrafish to understand the diversification paths of these two classes between jawless and jawed lineages. As zebrafish AC types have not yet been well characterized, we excluded them from the comparison.

In addition to four SAC types, we identified 11 non-SAC AC types in the lamprey retina (Fig. 1f, Supplementary Fig. 6a). Unlike jawed species (e.g. chicken, mouse) where most ACs can be classified as exclusively GABAergic or glycinergic based on transporter expression, lamprey ACs show a more complex pattern, with most lamprey ACs expressing both GABA (*SLC6A11*) and glycine (*SLC6A9*) transporters, suggesting a dual role (Supplementary Fig. 6b)[70]. AC8 is uniquely glycinergic with the expression of only the glycine transporter *SLC6A5* (*GLYT2*). AC1, AC6, AC7, AC9, AC10, and AC11 also express *SLC6A5* (*GLYT2*) (Supplementary Fig. 6b). These findings indicate that lamprey AC types can be distinguished by their variable glycine-transporter expressions and cannot be strictly categorized into GABAergic or glycinergic subclasses, with many cell types apparently utilizing both neurotransmitters.

We were surprised to discover that the number of AC types in the lamprey retina was less than one-quarter of AC types in chicken or mouse retina[19,34], indicating that lamprey ACs are less diversified compared to existing jawed species. When we compared lamprey ACs to mouse or chicken AC types, either through integration or transcriptomic mapping via XGBoost, we observed a consistent pattern: multiple mouse or chicken AC types corresponded to a single lamprey type with over 50% matching percentage (Fig. 5a-d). This correspondence suggests that AC types share similar origins between jawless and jawed lineages, but that these types may have further diversified into multiple sister types in jawed species. We also found shared orthologs in top matched types between the lamprey and mouse or chicken (Fig. 5e, f). Notably, the AII amacrine cell in the mouse retina (AC3) corresponds to lamprey AC8 with the highest confidence, sharing common markers including *CAR2* (also known as *CA2*) and *TNR* (Fig. 5c, e). However, the gap junction gene *GJD2* generally has low expression in lamprey ACs, and it is not expressed by AC8 but rather by AC5 (Supplementary Fig. 6a, c). We detected very little expression of any other lamprey gap junctional protein in AC8 (Supplementary Fig. 6c). These findings suggest that an AII-like AC might have already been present in jawless progenitors, but that these cells might not have had the same function or connectivity as in mammals. AC10 and AC11 in lamprey do not correspond to any types in mice, and AC11 also lacks corresponding types in chicken (Fig. 5c, d). AC11 may have evolved separately in lamprey or disappeared during the evolution of jawed vertebrates.

## The evolutionary diversification of RGC types
We identified 41 RGC types in the lamprey, a number similar to what has been reported in the chicken or mouse retina (Supplementary Fig. 7)[19,52,71]. Based on hierarchical clustering and the expression of shared marker genes, these 41 RGC types could be divided into seven subgroups: 1) *SEMA3A.1* and *LRFN4* positive RGCs, 2) *FOXP1* and *PRDM13* positive RGCs, 3) *TMEM121* positive RGCs, 4) *PIEZO2.1* and *PENK* positive

RGCs, 5) *VAT1* and *TAFA2.1* positive RGCs, 6) *OSBPL5* and *PDE11A* positive RGCs, and 7) *NOS1* and *NR4A2* positive RGCs (Supplementary Fig. 7). We also identified two intrinsically photosensitive retinal ganglion cells (ipRGCs), RGC5 and RGC13, with the highest expression of the melanopsin-like gene *OPN4l.1* (Supplementary Fig. 7)[63].

Morphological studies of lamprey RGCs have previously revealed two distinct subclasses, with 40% of RGCs located at the ganglion cell layer, and the remaining 60% located at the inner nuclear layer[72,73]. The distinct somatic position and axonal efferent routes of RGCs in lamprey suggest significant divergence between jawless and jawed RGCs (Fig. 1b). To understand the homologous relationship between lamprey RGCs and those in mouse, chicken, and zebrafish[19,52,74], we used transcriptomic integration and mapping methods. By comparing two integration methods–CCA and RPCA, we found that RPCA consistently yielded a higher number of corresponding cell types between lamprey and each jawed species with a lower entropy value in the confusion matrix derived by transcriptomic mapping via XGBoost (Fig. 6a, b, Supplementary Fig. 8). Thus, using RPCA integration, we identified 24 lamprey types that correspond to mouse, zebrafish, or chicken types (Fig. 6b, c). Among these, 11 types uniquely corresponded to mouse, three types to zebrafish, two types to chicken, and eight types shared among two or all three jawed species (Fig. 6c). These corresponding cell types shared multiple conserved markers (Supplementary Fig. 9). Of note, we found lamprey RGC5 and RGC13 both matched with mouse M1 ipRGCs (33_M1 and 40_M1dup)[75,76], confirming the conserved origin of ipRGCs between jawless and jawed lineages. Moreover, among the best matching types, we found that there are lamprey ganglion-cell types that correspond to W3 types (W3B and W3D), ON-OFF direction-selective ganglion cells (ooDSGCs), Tbr1-RGC, and F-RGC types (F-midiOFF and F-miniOFF) in mouse (Fig. 6b, c)[53,77–80]. From the known functions of their counterparts in the mouse retina, many of these conserved RGC types may encode retinal motion to direct eye movements[80].

There were, however, 17 lamprey RGC types that lacked reliable matches with any jawed species (match percentage <50%). To investigate distinguishing features of these 17 types, we categorized them as the "non-conserved" group and the remaining 24 types as the "conserved" group. Differentially expressed genes (DEGs) enriched in these two groups are primarily associated with distinct biological pathways, except for genes involved in the Gene Ontology (GO) term: Neuron projection (Fig. 6d)[81,82]. Genes assigned to this term are broadly involved in axonal and dendritic targeting. We found that different gene sets within this GO term are enriched in the two groups, suggesting distinct targeting patterns (Fig. 6e). Interestingly, the gene set enriched in the conserved group showed a significantly higher expression score in mouse RGCs compared to the gene set in the non-conserved group (Fig. 6f). This finding implies that lamprey-specific RGC types might display divergent axonal and dendritic targeting patterns compared to conserved RGC types, which may more closely resemble those of jawed species.

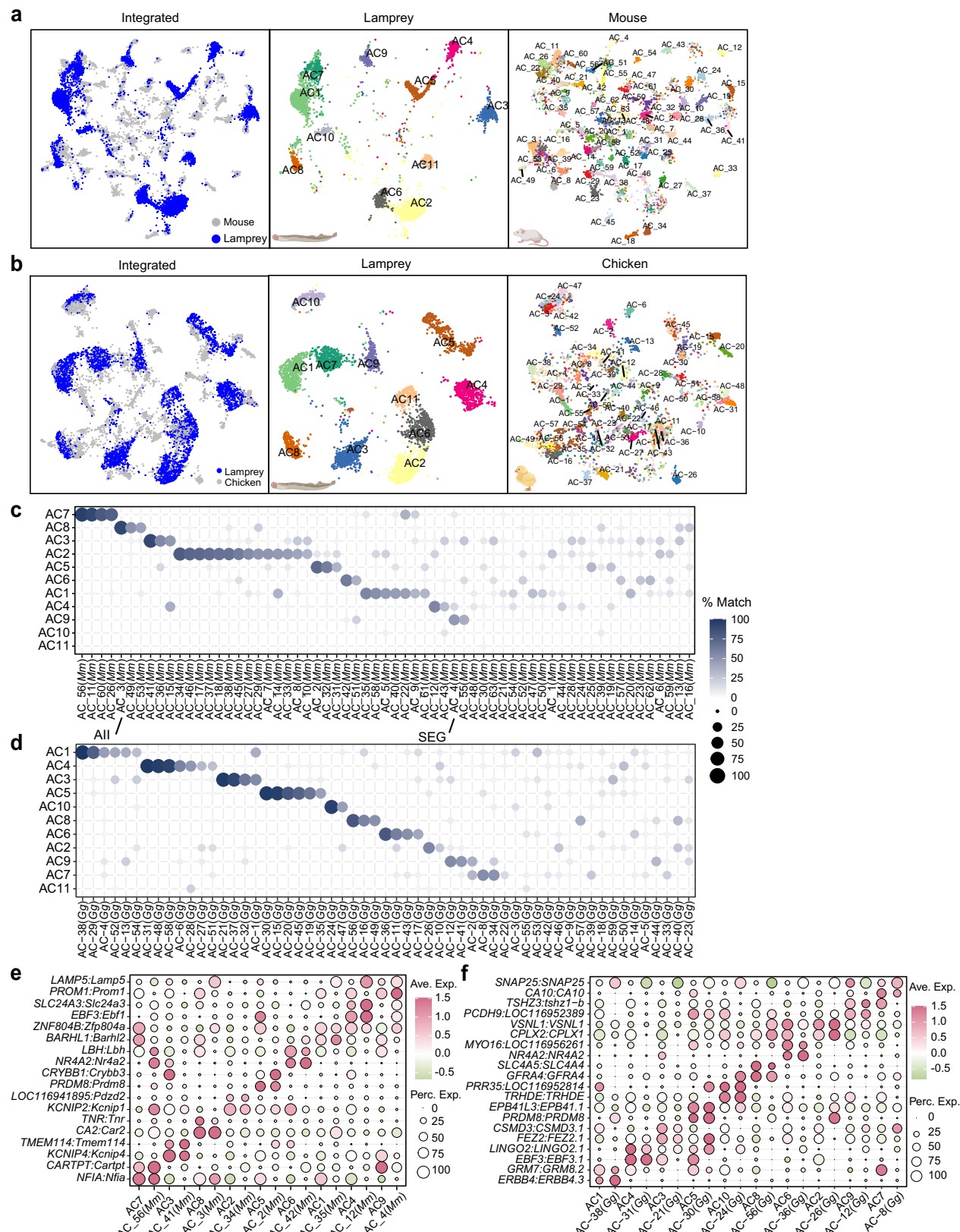

Altogether, these results indicate that the level of diversification of RGCs in the lamprey is similar to that in jawed species; however, over 40% of lamprey RGC types do not share a common origin with jawed species, suggesting extensive separate evolution in visual processing at the level of the ganglion cell.

**Ancient origin of genetic regulators in differentiating retinal cell classes**

The sharing of retinal cell classes across significant phylogenetic distances suggests that gene regulatory programs specifying these classes are also ancient, already emerging in vertebrate ancestors. To

**Fig. 5 | Classification of lamprey AC types and their comparison with jawed AC types. a** Integration of lamprey and mouse ACs visualized with UMAP, with both integrated and species-specific clusters presented in separated UMAP plots. **b** Integration of lamprey and chicken ACs visualized with UMAP, with both integrated and species-specific clusters presented in separated UMAP plots. **c** Confusion matrix demonstrating transcriptomic correspondence of AC types between lamprey and mouse. Lamprey ACs were used as the training dataset, while mouse ACs were used as the testing dataset. Mouse AII AC and SEG AC are indicated based on Ref 34. SEG ACs are glycinergic ACs, which are positive for Satb2, Ebf3,

and GlyT1 (Ref. 68). See Source Data. **d** Confusion matrix demonstrating transcriptomic correspondence of AC types between lamprey and chicken. Lamprey ACs were used as the training dataset, while chicken ACs were used as the testing dataset. See Source Data. **e** Dot plot showing expression patterns of conserved orthologous genes across corresponding AC types between lamprey and mouse. *CA2*, also known as *CAR2*. See Source Data. **f** Dot plot showing expression patterns of conserved orthologs across corresponding AC types between lamprey and chicken. See Source Data.

explore these ancient genetic regulators, we applied network-based regulon inference and protein activity analysis to compare the activities of essential regulators between the lamprey and three jawed species: chicken, mouse, and macaque (Supplementary Table 1)[19,30,34]. These regulators comprise transcription factors (TFs) and coTFs that drive the expression of cascades of downstream genes to specify the fate of cell classes[83,84]. Essential regulators also include signaling molecules and membrane proteins that influence molecular and physiological features of cell classes[5,85,86].

We used the Algorithm for the Reconstruction of Accurate Cellular Networks implemented with an Adaptive Partitioning strategy (ARACNe-AP) to infer the regulatory networks of candidate regulators[87]. We then developed Regulon Structure-based Activity inference (ROSA) to calculate their protein activities and infer essential regulators (Fig. 7a, see "Methods")[88]. Through this analytical framework, we identified all potential regulators in lamprey, chicken, mouse, and macaque datasets[19,30,34]. Based on the activities of these regulators, cells from each species were grouped into clusters, each corresponding to one of the six cell classes (Fig. 7b). We further identified highly active regulators specific to each cell class in each species (Supplementary Fig. 10). Many of these regulators, such as *ONECUT1, ONECUT2, VSX2*, and *SOX9*, have been shown to play a crucial role in the differentiation of retinal cell classes (Supplementary Fig. 10)[54,89–91]. These results demonstrated the precision of protein activity inference and confirmed the specificity of essential regulators for individual cell classes in these examined species.

We next compared active regulators present in individual cell classes between lamprey and each jawed species, assessing their levels of conserved usage between the two lineages. We first ranked these regulators based on their specificities in lamprey cell classes. We found that many essential regulators are also top active regulators in the corresponding cell classes in jawed species (Supplementary Fig. 11). Moreover, we identified multiple common regulators shared across these vertebrates (Fig. 7c). These regulators likely represent some of the most ancient genetic programs, possibly defining original cell classes in our vertebrate ancestors. We then inferred the level of conservation of cell classes between the lamprey and each jawed species by assessing the usage of conserved factors. Our criteria were based on the extent to which a similar set of essential regulators is shared with a similar usage between the lamprey and any jawed species. We used gene set enrichment analysis (GSEA) for this purpose and selected the top 50 regulators specific to each cell class in a jawed species as a gene set of interest. We calculated normalized enrichment scores (NES) and statistical p-values of this gene set referred to the lamprey cell-class signature (Fig. 7d) (see "Methods"). A higher enrichment score indicates a higher level of conservation, signifying that a similar set of essential regulators is shared between the two species. Our results showed that all cell classes demonstrated significant conservation, as highlighted by their p-values, with MG displaying the highest conservation and ACs the lowest (Fig. 7e). The relative lack of conservation of genetic regulators in ACs indicated considerable diversification between jawless and jawed lineages.

## Discussion

Considerable effort has recently been given to understanding the genetic characterization of different cell types in the nervous system.

Our study compared neuronal cell types between cyclostomes and other vertebrates, groups that diverged from one another in the late Cambrian over 500 million years ago[92]. We have shown that certain retinal cell types were clearly established in vertebrate progenitors at the time of the separation of cyclostomes, including rod and cone photoreceptors and certain retinal interneurons (such as starburst amacrine cells). Mechanisms of developmental regulation also appear to have emerged very early in the evolution of the retina.

A previously underestimated aspect of the lamprey retina is its rich variety of cell types, challenging the traditional understanding that early vertebrates possessed simpler nervous systems. We identified 74 different cell types, a number comparable to that in primates[30,93]. The molecular characteristics of these cell types and their parallels to jawed species highlight an impressive correspondence between lamprey and other vertebrates. There were, however, also some important differences. Lamprey retina has fewer amacrine cell types than mammals, and our results suggest that single types of lamprey amacrine cells are related to several types in mammals. Moreover, we found much less correspondence of ganglion cell types between lamprey and mammals than for the other cell classes. Our study has shown that retinal types during evolution were in some cases remarkably stable but in other cases underwent considerable divergence, probably reflecting the different behavioral and environmental constraints imposed on different species. Our work suggests that evolution proceeded opportunistically, preserving cells and circuits that maintained their usefulness, losing other cells and circuits that were no longer relevant, and inventing new mechanisms as these became adaptive.

### The origin of rod bipolar cells

Our work provides insights into the origin of the rod pathway[94,95]. A duplex retina denotes a retina consisting of rods and cones, which together mediate scotopic and photopic vision[38,39]. In the mammalian retina, a designated rod pathway emerged for transducing rod signals[96]. In this pathway, rods primarily contact a single type of rod bipolar cell (RBC). RBCs do not synapse directly with RGCs but instead terminate on a unique AC type, the AII AC[97]. Such an adaptation is believed to enhance vision in low-light conditions by allowing rod signals to be filtered and pooled when few rods absorb photons, and by then integrating these signals into cone pathways[98]. The AII ACs form gap junctions and inhibitory synapses with ON and OFF CBCs, which in turn connect to RGCs[57,99]. This indirect mechanism of rod signaling was once thought to be exclusive to mammals, but recent molecular and connectivity characterizations of the rod pathway in zebrafish have indicated that at least some lower vertebrates may also process rod signals in this fashion[35].

Our findings suggest that some mechanisms of rod signaling may have originated before the split of cyclostomes from other vertebrate ancestors. We have shown that rods are transcriptomically different from cones and that they use distinct phototransduction gene sets (Fig. 2a, b). Our research has also identified RBCs in the lamprey retina (Fig. 3). Lamprey RBCs express the marker genes *PRKCA/PRKACA* and *GRAMD1B* and bear a morphological resemblance to mammalian RBCs (Fig. 3b, d). Interestingly, transcriptomic mapping via XGBoost indicates that there are two ON RBC types in the lamprey closely correlated

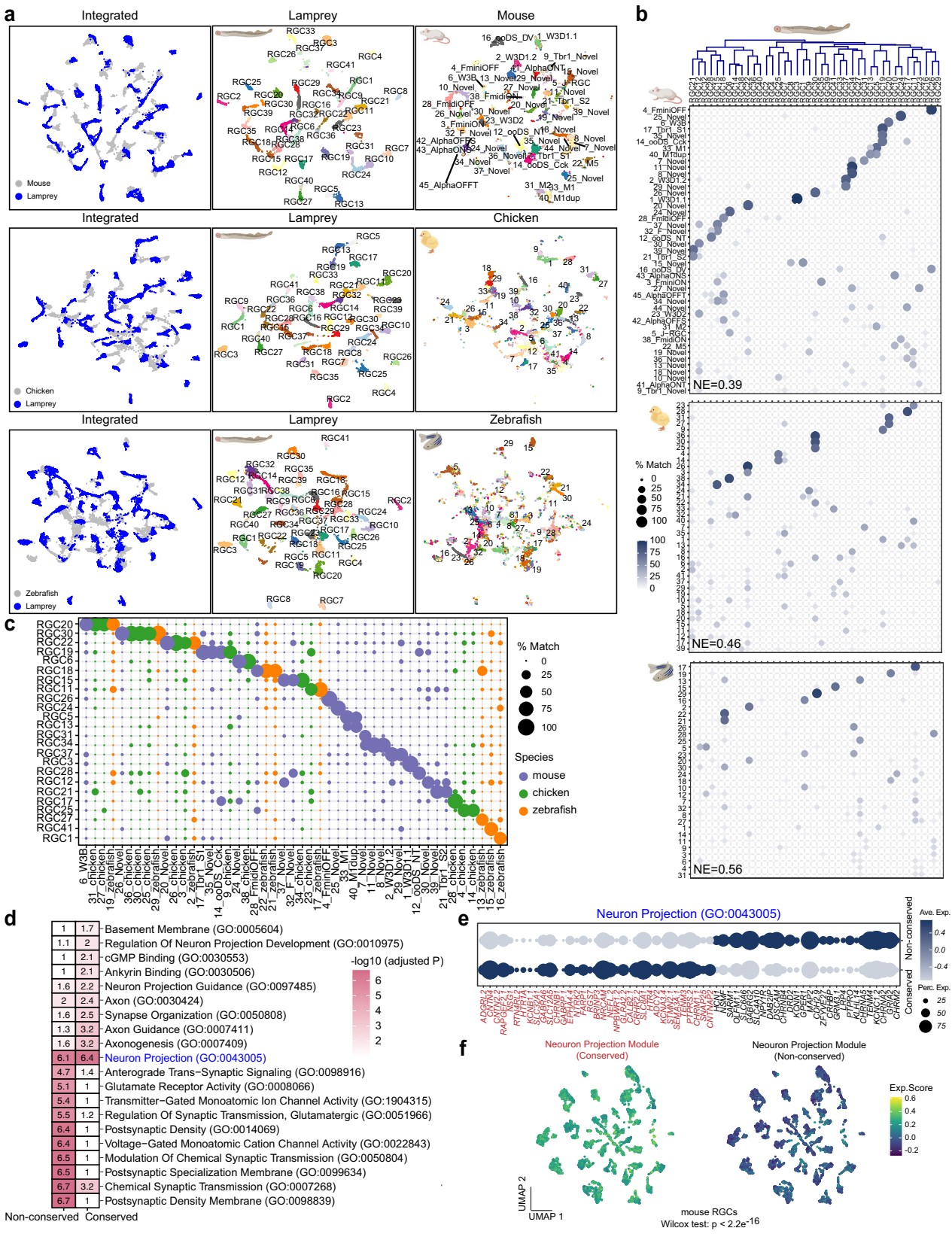

to RBCs found in zebrafish, chicken and mouse (Fig. 3f–i)[35]. Lastly, a critical component of the mammalian rod pathway, the AII-AC, is closely aligned to lamprey AC8 (Fig. 5c), though the AC8 cells do not express gap junction genes (Supplementary Fig. 6c).

These findings suggest that lamprey have a more primitive pathway for rod signaling, instead of or in addition to the AII-amacrine

pathway now utilized by mammals. Rods appeared during vertebrate evolution after cones[100] and may have initially utilized bipolar cells of both ON and OFF types similar to ON and OFF cone bipolar cells, directly contacting ganglion cells. It is significant in this regard that lamprey has been shown to have a purely rod bipolar cell which is OFF or hyperpolarizing[56]. Our observations also indicate that there is an

**Fig. 6 | Comparative analysis between lamprey and jawed RGC types.**
**a** Integration of lamprey RGCs with those of mouse, chicken and zebrafish visualized with UMAP, with both integrated and species-specific clusters presented in separated UMAP plots. **b** Confusion matrix demonstrating transcriptomic correspondence of RGC types in mouse, chicken, or zebrafish to lamprey RGC types. Lamprey RGCs were used as the training dataset, while RGCs from jawed species were used as testing datasets. Mouse RGC types follow the nomenclature in Ref. 52. NE, normalized entropy, was calculated for each confusion matrix. See Source Data. **c** Dot plot summarizing the matched RGC types between lamprey and the jawed

species of mouse, chicken, and zebrafish. See Source Data. **d** Enriched GO terms in conserved and non-conserved groups of lamprey RGCs. Hypergeometric tests are used. P-values adjusted by the false discovery rate are shown. **e** Distinct gene expression patterns in the GO term: Neuron Projection between the conserved and non-conserved groups. See Source Data. **f** Mouse RGCs showed a significantly higher expression score when using the module genes from the conserved group compared to module genes from the non-conserved group. A one-sided Wilcoxon rank-sum test was used. See Source Data.

OFF-BC type, BC7, which expresses the RBC marker *GRAMD1B* and may be the OFF RBC previously identified[56]. Moreover, OFF RBCs have been detected in the retinas of dogfish and amphibians[101–104]. These findings suggest that ON and OFF rod bipolar cells were present in lamprey progenitors before the split of cyclostomes from other vertebrate lineages. OFF RBCs may then have slowly disappeared in some lineages during evolution, as this earlier direct bipolar-to-ganglion cell pathway was replaced by the AII pathway now found in mammals and some other vertebrate species[95]. Future investigation into the morphology, physiology, and connectivity of lamprey bipolar and amacrine cells will be crucial for exploring these hypotheses and determining how rod pathways in lamprey resemble or differ from those in other vertebrates.

## ON and OFF pathways

Another notable feature of the lamprey retina is the presence of ON and OFF pathways, which discern the increment and decrement of light. The distinction between ON and OFF begins at synaptic connections between photoreceptors and bipolar cells in the outer plexiform layer (Fig. 1b), where BCs can be categorized into ON and OFF subclasses[105]. This functional differentiation is due to opposing responses to glutamate. In the dark, glutamate released by PRs depolarizes OFF BCs by activating their ionotropic glutamate receptors[50]. In contrast, photoreceptors hyperpolarize ON BCs via metabotropic glutamate receptors, which close non-selective cation channels now known to be transient-receptor-potential melastatin (TRPM) channels[106]. In mouse retina, OFF BCs can express the kainate receptor *Grik1*, while ON BCs typically express *Grm6* and *Trpm1*[107]. A previous study using physiological recordings demonstrated the presence of ON and OFF BC types in the lamprey[108]. This study showed that the light responses of ON BC are sensitive to AP4, suggesting that lamprey ON BCs express group III metabotropic glutamate receptors. Our results align with this functional result and have further revealed that lamprey ON BCs express *GRM8.2* (Fig. 3b, Supplementary Fig. 4a). Furthermore, it seems that *GRM8.2* associates with *TRPM3.1* in lamprey ON BCs (Fig. 3b, Supplementary Fig. 4a). Notably, *GRM8.2* and *TRPM3.1* are the orthologous genes of *Grm6* and *Trpm1* expressed by bipolar cells in jawed species (Supplementary Fig. 4e, f, Supplementary Data 2). Additionally, lamprey OFF BCs express *GRIK2*, an ortholog of the OFF BC gene *Girk1* in jawed species (Supplementary Fig. 4d). Thus, our results have confirmed that BCs in the lamprey retina differentiate into ON and OFF subclasses and have also identified the conserved usage of glutamate receptor genes in ON and OFF pathways.

An ON/OFF distinction extends into the inner plexiform layer (IPL) via specific synaptic interactions between presynaptic ON/OFF BCs and postsynaptic ON/OFF types of amacrine cells and retinal ganglion cells. Starburst amacrine cells (SACs) are among the earliest cell types to stratify their dendrites in the IPL and can be classified into ON and OFF types[109]. ON and OFF SACs have been shown to offer scaffolding for the innervation of respective BC types and have a pivotal role in organizing ON and OFF circuitry in the mouse retina[109,110]. Our study not only detected ON and OFF SACs in the lamprey retina but also observed similar gene expression patterns to those in mouse (Fig. 4g–i). Specifically, *FEZF2* is expressed by lamprey ON SACs, echoing the expression pattern of its mouse paralog, *Fezf1*, which

determines the fate choice of ON versus OFF SACs (Fig. 4i)[32]. Intriguingly, transmembrane proteins such as *TENM3* are also expressed differentially between lamprey ON and OFF SACs (Fig. 4l). These results suggest that a potentially analogous molecular mechanism governs ON and OFF laminations in the lamprey IPL. Additionally, *MEGF10*, which regulates the spatial arrangement of SACs in the mouse retina, is also present in lamprey SACs[68]. Altogether, the existence and division of SACs into ON and OFF subgroups seems to be a primitive and conserved trait across all vertebrates.

## Novel features of the lamprey retina and evolutionary modifications

The lamprey retina possesses several novel and distinguishing features. First, four types of horizontal cells (HCs) identified in the sea lamprey retina are localized at two distinct layers between the outer and inner nuclear layers (Fig. 4b)[40,41]. Although these cells are related to Type I and II HCs in chicken, their somatic positions are different from chicken HCs. As HCs are known to provide inhibitory feedback to rods and cones[111,112], the circuit connectivity and physiological function of these diverse HC types in lamprey merits future exploration.

Second, nearly all lamprey AC types express both GABAergic and glycinergic transporters, suggesting that lamprey ACs can utilize both of these inhibitory transmitters (Supplementary Fig. 6a, b). This is unlike other vertebrates, where most AC types have been shown to be either GABAergic or glycinergic but not both[34,113]. Lamprey AC types can be further distinguished by the expression of one or two different glycinergic transporters (Supplementary Fig. 6b). Moreover, each lamprey AC type corresponds to multiple types of mouse or chicken ACs (Fig. 5c, d). These results may be explained by the theory of "apomere" evolution of new cell types via module divergence[26]. In this theory, ancestral AC types could be multifunctional with multiple modules, and the diversification of AC types in jawed species may have occurred through the segregation of functions or modules into distinct sister AC varieties[26,114].

Lastly, RGCs in the lamprey retina feature the greatest diversity among all the cell classes, with 41 distinct types. This count is comparable to the number of RGC types in mouse or chicken retina. However, there is a stark contrast in terms of conserved RGC types between lamprey and jawed species, which may reflect a major difference in the localization of the somatic and axonal layer of lamprey RGCs compared to jawed species (Fig. 1b). Jones et al.[72] and Fletcher et al.[73] identified six morphological RGC groups in lampreys: two inner ganglion cell (IGC) groups, IGCa and IGCb; three outer ganglion cell (OGC) groups, OGCa, OGCb, and OGCc; and one bipolar ganglion cell group, BPGC. Of these, four (IGCa, IGCb, OGCa, OGCb) are homologous to RGCs in other vertebrates. These cells stratify their dendrites in the IPL and project to the tectum, directing eye, head, and body movements. Intriguingly, our results show that a couple of RGC types in the lamprey align with several direction-selective ganglion cells (DSGCs) including ON-OFF DGSCs and local motion detector W3 cells in the mouse retina (Fig. 6a–c)[53,80]. These results support the role of lamprey RGCs in mediating phototaxis[115]. Moreover, since DSGCs connect with SACs, these correspondences may reflect conserved synaptic partnerships with SACs. Direction-selective circuits for motion detection may have been one of the earliest design features of

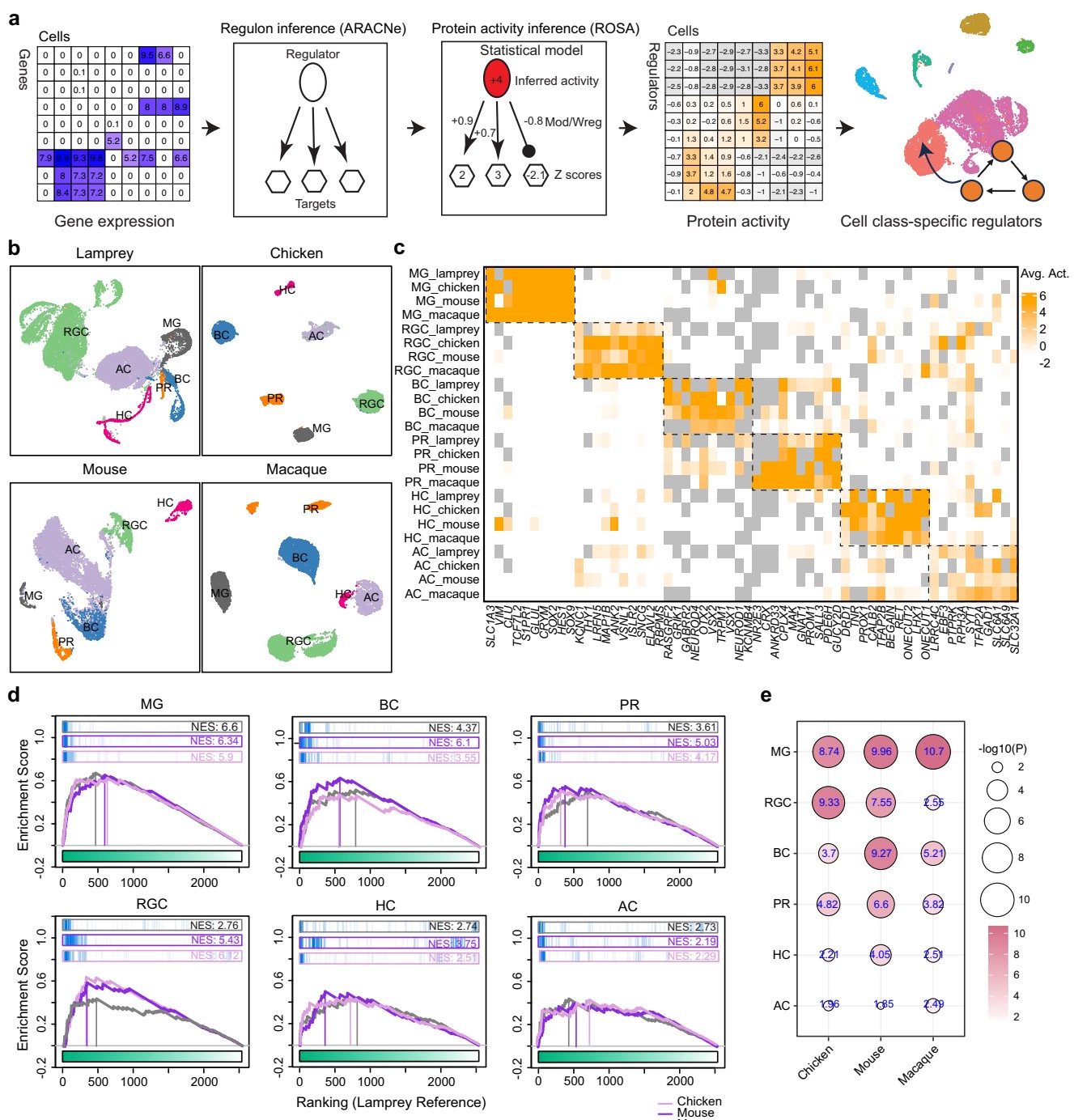

**Fig. 7 | Identification and comparison of protein activities of class-specific master regulators between lamprey and three jawed species. a** Analysis framework for network-based master regulator and protein activity inference. First, the gene regulatory network was reverse-engineered from the gene expression profile with ARACNe-AP. ROSA was then used to infer protein activity based on the regulon structure and gene expression status of the targets. Dimension-reduction analysis was conducted, and essential regulators, including TF, coTF, and surface/signaling proteins, were inferred from protein activity. See Methods. **b** Cells from lamprey, chicken, mouse, and macaque were clustered based on protein activities of master regulators and visualized with UMAP. Six cell classes, defined by the scRNA-seq analysis, were also separated in the UMAP. **c** Heatmap showing the protein activities of class-specific master regulators shared among lamprey,

chicken, mouse, and macaque. **d** GSEA plots showing comparisons of top active regulators between lamprey and each of the three jawed species in individual cell classes. Inferred top 50 regulators in each jawed species are used as the query gene set. The ranked cell-class signatures in the lamprey are used. Normalized Enrichment Score (NES) is calculated for each comparison. A higher NES indicates a higher degree of conservation. See Source Data. **e** Dot plot showing statistical p-values for each comparison between the lamprey and each jawed species in individual cell classes. The p-values were calculated based on a null model by permuting the samples uniformly at random 100,000 times. Two-sided tests were used. A higher -$\log_{10}$(p values) indicates a higher degree of significant conservation. All p values are smaller than 0.05 (-$\log_{10}$0.05 = 1.3). See Source Data.

the vertebrate camera eye, enabling gaze stabilization and object tracking[66]. Additionally, the correspondence of ipRGCs between lamprey and mouse suggests a Cambrian origin of ipRGCs and of the pupillary light reflex among vertebrates[116]. The remaining two subtypes (BPGC and OGCc) extend their dendrites into the OPL, directly contacting PRs and projecting to the pretectum, potentially regulating dorsal light and visual-escape responses. Our results identified a group of RGC types (which we called 'non-conserved'), which do not have a correspondence to ganglion cells in zebrafish, chicken, and mouse and may be peculiar to cyclostomes (Fig. 6). Interestingly, this subgroup might use distinct genes for axonal and dendritic targeting compared to the conserved RGCs shared with jawed species (Fig. 6d). It remains interesting to speculate whether this non-conserved subgroup of RGCs types was uniquely acquired by cyclostomes or was inherited from a vertebrate ancestor but lost in the evolutionary transition to jawed species. It is also possible that these ganglion cells evolved independently in lamprey as later adaptations to the visual behavior of this species.

In this study, we also emphasize a method of inferring regulatory protein activity from scRNA-seq datasets. While scRNA-seq provides a genome-wide profiling of gene expression in individual cells, the expression level of a gene may not always correspond to its protein activity due to post-translational modification[117]. Moreover, transcription factors involved in cell-type specification during development may have reduced expression in adult cells. Given the pleiotropic nature of transcriptional regulation and the evolutionary changes in the co-regulatory complex of transcription factors[26], we used network-based methods (Fig. 7). These methods consider the regulatory structure–that is, the relationship between gene expression of regulators and their targets, as well as the expression status of target genes to infer the activity of master regulators. With this approach, our method identified ancient regulators that might have originated in common vertebrate ancestors. Our study also suggests that the class specification of ACs may be the least conserved among all retinal classes between lamprey and jawed species (Fig. 7d, e). This finding, combined with the fewer number of AC types in the lamprey, could highlight a distinction in the generation and diversification of ACs between cyclostomes and other vertebrates.

## Limitations of our study
Despite the great diversity of cell types identified in this study, additional cell types could likely be revealed with the sequencing of more cells, including additional types among bipolar cells. Our characterization of cell types in the lamprey is primarily based on transcriptomic distinctions. Many of the cell types identified in this manner warrant future histological and function validation. In particular, the identification of a wide range of RGC types, along with future clarification of their localization, dendritic projections, physiological responses, and brain innervation, will provide important insight into understanding the diversification of RGC types in vertebrates.

From the improved gene annotation of the lamprey genome, we were able to project the best corresponding cell types between lamprey and jawed species using integration and transcriptomic mapping via XGBoost methods. The purpose of this survey is to provide a preliminary examination of ancestral cell types that might have emerged before the divergence of the cyclostomes from other vertebrate lineages. However, differences in the quality of genome annotation and the complexity of published scRNA-seq datasets might limit our findings.

## Methods
### Tissue procurement
Lamprey tissue collection was carried out in accordance with the recommendations of the Guide for the Care and Use of Laboratory Animals of the National Institutes of Health, USA, and was approved by the University of California, Los Angeles, Animal Research Committee.

Sea lamprey, *Petromyzon marinus* Linnaeus 1758, were provided by the Hammond Bay Biological Station of the United States Geological Survey (USGS), Millersburg, MI, USA. They were kept in a large fresh-water aquarium at 4 °C on a 12 h:12 h light:dark cycle. The lampreys used in the study were adults, which were post-metamorphic and sexually mature. Lampreys were deeply anesthetized with 400 mg/L tricaine methanesulfonate (MS-222; E10521, Sigma–Aldrich, St. Louis, MO) and decapitated. After dissecting out eyes, the anterior chamber and the vitreous were removed by rapid hemisection. The posterior eyecup was immersed in room-temperature Ames' medium (Sigma), equilibrated with 95% $O_2$/5% $CO_2$ for at least 20 minutes for cell dissociation, or immediately fixed with 4% PFA for immunohistochemical experiments or fluorescence in situ hybridization.

### Fluorescence in situ hybridization (FISH) validations
After isolation of the tissue, the posterior poles containing the retina were fixed with 4% PFA for 1 hour at 4 °C, rinsed with PBS and immersed in 30% sucrose at 4 °C overnight, and then embedded with Tissue Freezing Medium. The tissue was sectioned at 20-μm thickness and stored at −80 °C for long-term storage. Fluorescence in situ probes against specific lamprey genes were generated with previously described methods[30]. Briefly, total RNA was extracted from lamprey retinas and converted to cDNA libraries through reverse transcription with the AzuraQuant cDNA synthesis kit. Antisense probe templates for individual target genes were PCR-amplified from the cDNA libraries with a reverse primer having a T7 sequence adapter to permit in vitro transcription. DIG rUTP (Roche) and Fluorescein rUTP (Roche) were used to synthesize probes for single or double FISH experiments. Retinal sections were thawed, treated with 1.5 μg/mL of proteinase K (NEB) for 5 minutes, post-fixed with PFA for 5 minutes, and deacetylated with acetic anhydride for 10 minutes. After blocking, the retinal sections were incubated with one or two probes overnight. For single probe detection, the retinal section was incubated with anti-DIG HRP (1:1000, Roche) or anti-Fluorescein POD antibody (1:1000, Roche) overnight and fluorescent color then developed with tyramide signal amplification (TSA)[118]. A sequential probe detection procedure was applied for double fluorescence in situ hybridization. Here, after incubation with anti-DIG HRP antibody overnight and the completion of TSA for developing the first fluorescence color for the first probe, the HRP activity was quenched with 3% $H_2O_2$ for 30 minutes at room temperature. The section was then incubated overnight with anti-Fluorescein POD antibody for the second probe and followed with TSA reaction to develop a second fluorescence color. All RNA probes used in the study have been summarized in Supplementary Table 2.

### Immunohistochemistry
Tissue was fixed and prepared as described above. Mouse anti-PKCa (1:2000, Abcam # ab31, MC5) antibody was used in this study. Donkey anti-mouse secondary antibody, Alexa Fluor 488 (Jackson Immunoresearch #715-545-150) was used at 1:1000. Nuclei were labeled with DAPI (1:1000, Invitrogen #D1306). Sections were mounted in ProLong Gold Antifade (Invitrogen #P36930).

### Image acquisition, processing, and analysis
Images were acquired on an Olympus FluoView™ FV1000 confocal microscope with 405, 488, and 599 nm lasers and scanned with a 40X or 60X oil objective at a resolution of 1024×1024 pixels, a step size of 1 μm, and an 80 μm pinhole size. Maximum intensity projections were generated with ImageJ[119] (NIH) software, and brightness and contrast adjustments were made with Adobe Photoshop.

### Construction of a retina-specific transcriptome for *Petromyzon marinus*
Lamprey retinas were separated from the choroid and the rest of the eye cup and were immediately placed in RNA*later*™ Stabilization

Solution. RNA was extracted from the stabilized retinal tissues with RNeasy Plus Universal Mini kit (QIAGEN) by following the manufacturer's instructions. We obtained high quality total RNA from lamprey retinal tissue [RNA Integrity Number (RIN) score: 9.8] and prepared strand-specific libraries with the TruSeq strand-specific Total RNA kit (Illumina Inc.). The resulting RNA was sequenced on the NextSeq 500 system to obtain 50 million 100 bp paired end reads. We merged TruSeq bam files of samples S1 and S2 by "samtools merge"[120] and generated a merged fastq file with "samtools bam2fq". We then updated the gtf file of NCBI genome assembly kPetMar1.pri (https://www.ncbi.nlm.nih.gov/datasets/genome/GCF_010993605.1/) of sea lamprey (*Petromyzon marinus*) according to the following steps. (1) We used HISAT2[121] for reads alignment. We used the hisat2_extract_splice_sites.py and hisat2_extract_exons.py scripts to extract splicing sites and exons. We then used "hisat2-build" to create hisat2 index. The final bam file was then generated by hisat2. (2) We used StringTie[122] for transcript assembly and quantification. The gtf file was updated by stringtie by using the HISAT2 bam file. The stringtie gtf and the reference gtf were compared with gffcompare[123] and merged with "stringtie−merge". Transcript abundances were estimated with stringtie (with arguments -e -B). After the update, the NCBI+TruSeq reference transcriptome had 15,070 genes updated by StringTie with MSTRG Number. We used the updated transcriptome for aligning scRNA-seq reads (see below).

The NCBI reference genome of the sea lamprey (kPetMar1.pri) has over 60% of its genes assigned with a LOC number, indicating that these genes have uncertain functions. To facilitate cell type classification using canonical retina markers, we further annotated these genes based on the homologous relationship of their protein sequences with those of human and other species. Specifically, the following steps were undertaken. First, we converted the LOC gene IDs to RefSeq protein IDs using bioDBnet (https://biodbnet-abcc.ncifcrf.gov/db/db2db.php). We then retrieved the corresponding fasta sequences in batch using efetch (Entrez Direct E-utilities). Next, we used the command line version of blastp from BLAST+ to find their target sequences in humans, using the arguments '-remote -db nr -entrez_query "Homo sapiens [organism]".' We then retrieved the top aligned human RefSeq protein ID for each query sequence and converted it to the corresponding human gene symbols using Entrez Direct E-utilities (esearch, elink, esummary and xtract). Following this, we annotated 8984 LOC genes with the format for human gene symbols, which are written entirely in uppercase. For 274 LOC genes without a significant match in humans, we used a similar strategy using blastp to search for matches in other species. These genes were annotated in lowercase. We also labeled genes sharing the same protein by adding a suffix to the gene name (Supplementary Data 1). With this method, we improved the proportion of annotated genes in the reference transcriptome from 35.86% to 68.92% (Supplementary Fig. 1c).

### Single cell RNA-sequencing library preparation
Adult lamprey retinas were dissected by separating them from the choroid and the rest of the eye cup. The retinas were digested with papain (40 U) at 37 °C for 15 minutes and then dissociated into a single-cell suspension for single-cell RNA sequencing (scRNA-seq) with a targeted cell number of 10,000 cells using the Chromium™ Next GEM Single Cell, 3' Kit v3.1 (10X Genomics). cDNA was amplified with 11 cycles, and Libraries were sequenced on the Illumina NovaSeq S4 platform.

### Single-cell RNA-seq data analysis
**Read alignment and generation of count matrices.** We used the updated "NCBI+TruSeq" transcriptome to align scRNA-seq reads with Cellranger (10x Genomics, version 7.0.1)[124]. We first constructed a reference transcriptome with "cellranger mkref". We then mapped scRNA-seq reads with "cellranger count". The resulting count matrices have gene IDs from the TruSeq update with MSTRG number or LOC number. We further updated count matrices by replacing gene IDs, which were either MSTRG or LOC numbers, with annotated gene symbols following Supplementary Data 1.

**Data pre-processing, normalization, dimension reduction, and clustering analysis.** The raw filtered count matrices were used for further analysis with the Seurat R package (version 4.3.0)[125]. We first removed low-quality cells (gene counts < 500 and feature counts < 330) and putative doublets (feature counts > 8950) by examining the distributions of numbers of expressed genes and RNA counts detected in individual cells. Data normalization and identification of highly variable genes (HVGs) were performed with the SCTransform() function. A Gamma-Poisson generalized linear regression (glmGamPoi) model was used for the normalization[126]. We then performed principal component analysis (PCA) with HVGs and further eliminated batch effects with Harmony[127]. After Harmony correction of the top 50 PCs, a K-nearest neighbor (KNN) graph was constructed with Harmony components, and cells were clustered with the Louvain algorithm. The uniform manifold approximation and projection (UMAP) dimension reduction was used for visualization.

**Retinal cell-class annotation.** We annotated lamprey retinal cell clusters into distinct classes with the reference-based method SingleR[128]. First, we generated the reference dataset by selecting a small subset of clusters, each of which showed strong specific expression of canonical marker genes for certain cell classes. The remaining clusters were designated as a query dataset. Then we used the $\log_2(\text{CPM}+1)$ transformed data as input for SingleR to make predictions, where CPM stands for counts per million. We assessed annotation by conducting a detailed examination of class-specific markers for each cluster. We also evaluated the clustering patterns of these classes in the UMAP space generated by changing the number of Harmony components used. We observed that when using the top 5 Harmony components, cells from the same class predominantly clustered together in the UMAP, indicating the accuracy of the final annotation.

**Cell-type classification within each class.** To resolve cellular heterogeneity at a higher resolution, we divided cells into individual classes and further clustered cell types within each class. We performed data normalization with SCTransform(), PCA analysis, batch correlation with Harmony, and clustering as described above. We constructed the KNN graph using the first 50 Harmony components with the function FindNeighbors() and performed modularity optimization with the FindClusters() function. We visualized the separated cluster patterns using UMAP to ensure consistency with the assigned clusters, confirming that no over-clustering or under-clustering occurred. After initial clustering, we assessed cluster quality by evaluating the number of expressed genes, RNA counts, percentages of mitochondrial genes, and the expression level of cell-class marker genes. Low-quality cells and clusters with contaminated cells were removed, and a repeat round of clustering analysis was performed until every cluster contained high-quality cells. Parameters for filtering out low-quality cells are provided in Supplementary Table 3.

### Hierarchical clustering analysis
The correlation distance is defined as

$$Dist = \frac{1}{2}(1 - cor), \tag{1}$$

where *Dist* refers to distance and *cor* refers to the Pearson correlation. To generate the dendrogram, the hclust() function in the stats R

package with centroid agglomeration was used. We also used the neighbor-joining (NJ) method to generate the phylogenetic tree for all cell types across six classes. Bootstrap was performed by using ape[129] with 100 replicates generated. The CompexHeatmap[130] R package was used for the heatmap visualization of the correlation coefficients.

## Ortholog inference with OrthoFinder

We determined orthogroups and orthologous genes with OrthoFinder[131], a tool that uses a tree-based phylogenetic approach. Peptide sequences for lamprey, chicken, and macaque were retrieved from NCBI, while those for zebrafish, mouse, and human were retrieved from Ensembl (http://www.ensembl.org/). To remove redundancy in the NCBI peptide sequence files, we used a custom-written R script to select the longest peptide per gene for further analysis. For sequences downloaded from Ensembl, we used the Python script ("primary_transcript.py") built into the OrthoFinder to extract the longest transcript per gene. Subsequently, OrthoFinder was run on these non-redundant peptides. Phylogenetic trees of interested ortholog groups, such as *GRIK*, *GRM*, and *TRPM*, were visualized via MEGA[132].

## Cross-species integration of the retinal cell classes

**Selection of most informative orthologous genes.** Before integrating cells between two species, we constructed ortholog groups between two compared species. Non-one-to-one (one-to-many, many-to-one, and many-to-many) orthologous relationships often arise due to various gene duplication events in evolution[62]. To address this complexity, we developed the Most Informative Orthologous Gene (MIOG) selection method (Supplementary Fig. 4c (ii)). After inferring the orthologous genes using OrthoFinder, we selected the MIOG from each ortholog group for each species. To do this, we first performed log normalization to standardize counts against the sequencing depth. We then calculated the standard deviations of the orthologous genes within the same orthologous group and selected the one with the highest values as the MIOG for integration.

**CCA integration.** We used canonical correlation integration (CCA) in Seurat[125,133] for cross-species integration. First, we normalized samples from each species by fitting the Gamma-Poisson generalized linear regression model implemented in the SCTransform() function. We then selected conserved HVGs with the SelectIntegrationFeatures() function. After finding anchors with the FindIntegrationAnchors() function with parameter dims=1:50, the datasets were integrated by using IntegrateData() with parameter dims=1:50. After integrating the data, we performed dimension reduction and clustering analysis with the top 50 principal components (PCs). To ensure a balanced representation across the diverse cell types, we downsampled cells from these groups. For lamprey and mouse RGC integration, we used the intersection method to select integration features. First, we identified mouse and lamprey HVGs by running the SelectIntegrationFeatures() function for each species. Then, we identified the shared conserved HVGs for these two species by intersecting the mouse HVGs and lamprey HVGs. After that, the datasets from the two species were integrated with IntegrateData(), followed by dimension reduction and clustering. The published datasets and detailed parameters for the cross-species CCA integration are summarized in Supplementary Table 1 and Supplementary Table 4, respectively.

**RPCA integration.** Seurat CCA may lead to over-integration when proportional cell types are not shared between species. To mitigate this, we also employed RPCA, which particularly has advantages when cell types differ substantially between datasets. Additionally, we fine-tuned the k.anchor parameter to enhance the integration strength, specifically addressing the challenges posed by gene expression shift. The integration procedures are the same as the CCA integration

described above with the intersection method used to select integration features. The published datasets and detailed parameters for the cross-species RPCA integration are summarized in Supplementary Table 1 and Supplementary Table 4, respectively.

## Transcriptomic mapping via XGBoost

To identify the correspondence of cell types across species, we performed a supervised multi-class classification using XGBoost[36] (R version), a scalable machine learning algorithm that uses the gradient tree boosting technique. Implementation comprised two steps: First, a predictive model was built from a training dataset from one species with labeled cell types; Second, the model was used for testing the dataset from another species to predict the corresponding cell-type label. The analysis details are outlined as follows. We first identified the common HVGs between species in the integrated assay with the SelectIntegrationFeatures() function in Seurat, and we used these genes as features in XGBoost. We constructed a predictive model using 67% of the cells from each cluster in the training dataset (with a maximum of 300 cells per cluster if the cluster contained more cells), and we used the remaining cells to evaluate the model's prediction performance. After the model achieved high performance, we applied the testing dataset to it. Finally, we used a confusion matrix to demonstrate the correspondence of cell types between species. A higher matched percentage indicates a higher similarity between cell types.

## Integration performance comparison using entropy analysis

We evaluated the performance of different integration methods from the confusion matrix, which is the matrix of predicted probabilities or classifications from the model. To do this, we used normalized entropy, a measure that quantifies the uncertainty of the prediction. Fewer random predictions result in smaller normalized entropy values. For a discrete random variable $X$ with probability mass function $p(x)$, the entropy $H$ is defined as

$$H(X) = -\sum_{i=1}^{n} p_i \log p_i \qquad (2)$$

We calculated entropy for each cell-type prediction and aggregated these values for each confusion matrix. The normalized entropy of the confusion matrix, ranging from 0 to 1, is given by

$$H_{norm}(X) = \frac{\sum_{j=1}^{m} H(X)_j}{\sum_j H_j^{max}} = \frac{-\sum_{j=1}^{m} \sum_{i=1}^{n} p_{ij} \log p_{ij}}{m \log(n)} \qquad (3)$$

where n and m represent the numbers of cell types in the training dataset and prediction dataset, respectively. Base e was used in the calculation.

## Detection of conserved markers across species and across samples

First, differentially expressed genes (DEGs) within individual species or samples were identified with the Wilcoxon test, as implemented by the "FindAllMarkers" function in Seurat. To eliminate batch effects from different species or samples, we selected the conserved makers with the Stouffer's integration method[134]. The individual p-values from the DEG tests were transformed into Z scores. Subsequently, these Z scores were combined by using the formula below:

$$Z_{Stouffer} = \frac{\sum_{i}^{n} Z_i}{\sqrt{n}} \qquad (4)$$

where $i$ is the species or sample index.

Furthermore, the Stouffer-integrated Z scores were converted into p-values under the assumption of a standard normal distribution, and they were then adjusted with the False Discovery Rate (FDR) method. To avoid infinity of Z scores, p values equal to 0 were replaced

by the smallest positive double-precision number in R during conversion.

## Gene ontology enrichment analysis

We identified significantly enriched Gene Ontology (GO) terms among different sets of differentially expressed genes (DEGs) across different cell types or groups. Gene set enrichment analysis was conducted using the enrichr() function from the "enrichR" R package[135]. We used the databases "GO Molecular Function 2023", "GO Cellular Component 2023", and "GO Biological Process 2023"[135]. The module score of the genes in the enriched GO terms was calculated via the AddModuleScore() function in Seurat.

## Inferring protein activity of genetic regulators

**TF, coTF, and surface/surface signaling gene assignment.** We used GO annotations for human transcription factors (TFs), transcription cofactors (coTFs), and surface/surface signaling genes to curate a list of candidate regulators. The list of potential regulators comprises 1,430 transcription factors (TFs), 2744 coTFs, and 2717 surface and surface-signaling genes. The GO terms associated with these genes are listed as follow. TFs: "DNA-binding transcription factor activity" (GO:0003700)[136]. coTF (co-Transcription Factors and chromatin remodeling enzymes): "transcription coregulator activity" (GO:0003712); "DNA binding" (GO:0003677); "transcription factor binding" (GO:0008134); "regulation of transcription; DNA-templated" (GO:0006355); "histone binding" (GO:0042393), and "chromatin organization" (GO:0000790) [10]. Overlapped genes belonging to the TFs were removed. Surface proteins and surface-signaling proteins: "surface proteins" (GO:0009986); cell-cell signaling "GO:0007267" and cell-surface receptor signaling pathway "GO:000716". Redundant genes across categories were removed.

We determined orthologous relationships between lamprey and human genes from two sources: NCBI annotation, where orthologous genes share the same gene symbols; and OrthoFinder results, including one-to-one, many-to-one, and many-to-many matched genes within orthoGroups. Lamprey TFs, coTFs, and surface/surface signaling genes were annotated based on the GO terms of their human orthologs. For TF, coTFs and surface/surface signaling genes in chicken, mouse and macaque, we converted gene symbols to their human orthologs using ENSEMBL orthologous relationships and selected candidate regulators based on the human reference list.

**Reverse engineering the gene regulatory networks.** The TF, coTF and surface/signaling proteins selected from each species were used to generate their regulatory networks. Regulatory networks were reverse-engineered with ARACNe-AP[87]. ARACNe-AP was run with default settings: bootstrap n = 200, mutual information threshold "P < 10⁻⁸" and DPI (data processing inequality; enable=yes). We generated networks for individual samples to avoid batch effects. To handle the imbalance of the dataset, we used the downsampling technique to ensure that a substantial number of cells was included from classes with a small cell population. The downsampling parameters are provided in Supplementary Table 5. We further used the VIPER[88] R package to refine regulons from the networks by pruning the regulons up to size 50 (up to 50 targets for each candidate regulator) or removing the regulons whose size was smaller than 20 for downstream protein-activity analysis.

**Regulon structure-based activity inference.** Protein activities were inferred using a newly developed algorithm, Regulon Structure-based Activity inference (ROSA, https://github.com/JunqiangWang/Rosa). ROSA is an algorithm to perform regulon structure-based activity analysis using a statistical model. ROSA calculates a normalized enrichment score (NES) representing the relative protein activities of the candidate regulators.

First, the enrichment score (ES) of the regulator is calculated by

$$ES = \sum_i mod_{reg}^i . w_{reg}^i . Z^i \qquad (5)$$

where $i$ indicates the $i^{th}$ target of the regulator, $mod_{reg}$ is the regulation mode, indicating positive or negative regulation, $w_{reg}$ is the weight of the regulatory strength, and $Z$ is the adjusted $Z$ score of gene expression based on a coarse-grained approach (see below).

Based on the null hypothesis that Z scores follow the standard normal distribution, we can derive that ES also follows a normal distribution with mean of 0 and variance equal to $\sum_i (mod_{reg}^i . w_{reg}^i)^2$.

The NES is then calculated by normalizing the ES as follows:

$$NES = \frac{ES}{SD_{ES}} = \frac{\sum_i mod_{reg}^i . w_{reg}^i . Z^i}{\sqrt{\sum_i (mod_{reg}^i . w_{reg}^i)^2}} \qquad (6)$$

**A coarse-grained approach to adjust Z scores in ROSA.** Z scores from scaled $\log_2(CPM + 1)$ or regression residuals can have extreme positive or negative values. To avoid the effects of extreme values, a coarse-grained adjustment is performed. We generated a sequential set of Z scores $\{Z_1, Z_2, \ldots, Z_q\}$ which are the q-quantiles $\{1/q, 2/q, \ldots, (q-1)/q\}$ of the random variables from a standard normal distribution. Thus,

$$\Pr[Z < Z_i] = \frac{q_i}{q} \qquad (7)$$

For an unadjusted Z score, the adjusted Z score is given by

$$Z_{adjusted} = \begin{cases} \min\{Z_i, |, Z_i \geq Z\} \ if \ Z < 0 \\ \max\{Z_i, |, Z_i \leq Z\} \ if \ Z \geq 0 \end{cases} \qquad (8)$$

The adjusted Z scores were used to calculate ES and NES.

## Clustering analysis of protein activity and differential activity analysis

We performed dimension reduction and clustering analysis of protein activities. We calculated standard deviations of protein activities and selected the top 700 proteins with the highest standard deviations as the features for PCA analysis. Then we performed PCA, batch correction with Harmony, and UMAP dimension reduction based on the top 50 PCs. Further, we performed differential protein-activity analysis with the Student's t-test to infer class-specific regulators. Customized R scripts were used to perform the analysis by implementing functions in Seurat.

## Gene set enrichment analysis (GSEA)

Gene set enrichment analysis was performed to estimate the conservation of essential regulators across species. Class-specific differential signatures were generated by calculating the mean difference of the protein's activity scores between one class and all the rest of classes for each species. Rank signatures were derived from differential signatures using rank transformation. The top 50 highly activated proteins in the signature for each jawed cell class were used to query how these proteins are distributed in the lamprey class-specific ranked signatures. The NESs and p-values were calculated based on a null model by permuting the samples uniformly and at random for 100,000 times[137].

## Reporting summary

Further information on research design is available in the Nature Portfolio Reporting Summary linked to this article.

## Data availability

The raw TruSeq data have been deposited to the Sequence Read Archive (SRA) with accession numbers SRX26477354 and SRX26477355. The scRNA-Seq data generated in the study have been deposited to the Gene Expression Omnibus (GEO) under accession ID GSE236005. Previously published data utilized in this paper were downloaded from GEO repositories with the following accession numbers: GSE135406 (mouse PR), GSE81905 (mouse BC), GSE159107 (chicken), GSE237214 (zebrafish BC), GSE132555 (mouse SAC), GSE149715 (mouse AC), GSE137400 (mouse RGC), GSE152842 (zebrafish RGC), GSE118480 (macaque). This information is also summarized in Supplementary Table 1. The genome annotation files have been deposited to Zenodo (https://doi.org/10.5281/zenodo.13975013)[138]. The analyzed objects for the lamprey and other species in this study have been deposited to Zenodo (https://doi.org/10.5281/zenodo.14032062)[139]. Fluorescence in situ images and immunohistology images have been deposited to Zenodo (https://doi.org/10.5281/zenodo.13988889)[140]. Icons for species in the Figs. 3, 5, 6 were obtained from BioRender.com (https://BioRender.com/m51n417)[141]. Source data are provided as a Source Data file. Source data are provided with this paper.

## Code availability

All scripts are available via the Github page (https://github.com/PengYRLab/LampreyRetinalCellAtlas) and have been deposited to Zenodo (https://doi.org/10.5281/zenodo.14043155)[142] and (https://doi.org/10.5281/zenodo.14042584)[143].

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

## Acknowledgements

This work was supported by a career development award from Research to Prevent Blindness (Y.R.P.), a career starter award and competitive renewal grants from Knights Templar Foundation (Y.R.P.), a Klingenstein-Simons Neuroscience Fellowship (Y.R.P.), NEI grants R01 EY035324 (Y.R.P.), R01 EY001844 (G.L.F.) and R01 EY029817 (A.P.S.), a grant from the Great Lakes Fishery Commission (G.L.F.), the UCLA-Caltech Medical Scientist Training Program (T32 GM008042; NIH, NIGMS) to M.C., an unrestricted grant to the Department of Ophthalmology from Research to Prevent Blindness, Inc., and Core Grant P30 EY00331 to the Jules Stein Eye Institute. The authors thank R. Ramarapu, T. Xu, C. He, and S. Jakab for their technical assistance.

## Author contributions

Y.R.P. conceived and supervised the study. J.W. and L.Z. performed the computational analysis with the assistant from J.S.; Y.R.P. performed TruSeq and scRNA-seq experiments. A.P. and M.C. conducted the histology and in situ experiments; A.M. collected and dissected tissues under the supervision of G.L.F. and A.P.S.; Y.R.P. and G.L.F. wrote the paper with input from other authors.

## Competing interests

The authors declare no competing interests.
