## [Transparent Peer Review file · Nature Communications]

Molecular Characterization of the Sea Lamprey Retina Illuminates the Evolutionary Origin of Retinal Cell Types

Corresponding Author: Dr Yi-Rong Peng

Version 0:

Reviewer comments:

Reviewer #1

(Remarks to the Author)

These authors have an authoritative perspective on the evolution and physiology of the lamprey retina. They are the experts. Here, they expand this field by re-investigating old & unresolved questions about the agnathan retina cell types. With that new data in hand, new ideas can be presented about the earliest vertebrate retinas. Considering the long history of fascination with eye evolution across centuries, which continues to this day, the current manuscript will be of broad interest. The work is completed with great care and the analysis of cell types and master regulators seems to be sound. Congratulations on another landmark contribution.

The authors have chosen to use mouse as the representative gnathostome for most of their analyses. This is unfortunate because mice (like all mammals) have a retina that has gone through dramatic regressive phases of evolution and no longer has the characters of a typical gnathostome retina. Happily, this is not causing the authors too many issues of interpretation for many of their comparisons, e.g. where the authors are pointing out similarities between cell types in lampreys and gnathostome (e.g. the RBCs, the PR). However it creates a large challenge of interpreting the situation when noting that the cell types do not align well – e.g. The abstract reads “...conserved cell types shared between jawless and jawed lineages [are impressive]... In contrast to this evidence for conservation, the pathways of diversification for amacrine cells and retinal ganglion cells appear to have distinctly diverged between the two lineages.” A more likely interpretation is that mammals (especially the nocturnal clinically blind mice) have diverged away for other gnathostomes as they regressed into blindness. Two good solutions to this would be to i) alter the wording to not suggest mice are an exemplar of gnathostomes [e.g. in the abstract the authors are comparing lamprey vs. mice, not jawed vs. jawless]; ii) compare the lamprey data to chicken or frogs or fish.

I think this paucity of comparison to non-mammalian gnathostomes, if it remains in the manuscript, warrants comment in the section “limitations of our study”.

A paper by Hahn et al, cited by the authors as reference 21 and a preprint in BioRxiv, is now published in Nature, <https://doi.org/10.1038/s41586-023-06638-9>, and largely contains similar conclusions to the current submission but with 17 species compared rather than a focus on the lamprey. I welcome the focus on lamprey as a window into appreciating the evolutionary origins of the vertebrate retina, and so I think the current contribution will be impactful on the field. Some text, e.g. line 66, could be rephrased to reflect this recent work.

The authors provide useful validation of their RNA-Seq data by including in situ hybridizations. Some further details would make the latter a lot more convincing: a Supplemental Table with details about the riboprobes used (primers, probe length). The authors do a very good job of convincing the riboprobes are not suffering technical artefacts (e.g. the anti-DIG-HRP and Anti-Fir-POD are actually being very well differentiated, despite both being peroxidase), but they do not really communicate this in their Methods or Results – Supplemental Fig 3 shows great experimental care.

The manuscript would be strengthened by detailing if the key gene comparisons are orthologous. E.g. a synteny analysis for key genes that discriminate cell types would be appropriate.

Minor ideas: (I read the title & abstract and then looked at the Figs, as is typical, so some comments may be moot once the reader digs into the main text).

a) Fig 1b the schematics are intuitive to me as a retina aficionado, but I wonder if there is space in the legend to define the

grey blobs (nuclei).

- b) Fig 1e suggest labelling the ONL, INL, etc.. I'm not 100% sure what I am looking at... Same for Fig S3 (looking at S3 makes me even less sure about what layers Fig 1e is showing, but with some labelling of layers I think you could have it accomplish the opposite).
- c) Figure 3: panel c (FISH) is accidentally labelled with a "b"
- d) The order of ideas in line 53-57 could be more pleasing – rods vs. cones is spread out among intervening thoughts.
- e) Where is the RPE? And I'm always confused by retina scRNA-seq papers – aren't there any blood cells or microglia? If these were ignored, then that is very reasonable but could be stated in the Methods or perhaps at the start of the Results (that begins with an overview of the authors' approach)
- f) Similarly, the interpretation is centered on differentiated cell types – were cells in earlier stage of development excluded?
- g) The first section of the Methods "Tissue Procurement" would benefit from some additional details. E.g. these were post-metamorphic lamprey? Sexually mature? What does "posterior eyecup" include [sclera, RPE, choroid]?
- h) The writing on line 360 could be reworked to include the concept of regression. Much of the differences between lamprey and mouse (or lamprey and macaque) must be driven by the nocturnal bottleneck and loss of retinal complexity at the emergence of the mammals. I'm only encouraging a brief acknowledgment, e.g. some cells were lost while others were preserved....
- i) Line 365 is a bit overstated (the whole section is a bit discombobulated but the connections of ideas are appreciated). Your data speak to cell types being present, and so you can say that the cell substrates that underpin rod circuitry are there, but you cannot say much yet about circuitry. You've also almost ignored cone circuitry in this section (which is fine but then it is not about duplex retina), and the writing is framed in a way that fails to acknowledge that mammal rod circuitry is not a good representation of gnathostomes (so it cannot be used to infer the origins!). Kudos for corrective writing that acknowledges some of the complexity beginning on line 391.
- j) Line 452 typo? Mouse has two (smudgy and regressed) cone types.
- k) Line 448 I agree that retaining four HC types in lamprey is fascinating. How does one decide that there is only one type of cone in lamprey? E.g. there must be two or three subtypes within the cones you detected (even if they all share only one opsin type)? No need to address this silly point, but I don't really understand where one decides to be a lumpers or splitter among these (or any) cell types – surely not all the cones you detected were identical....
- l) Line 457 again you assume that mouse retina is more advanced than lamprey retina. Evolution is not always so progressive, and mice have lost lots compared to fish and birds. But you are just speculating about the history here, so I guess it is ok.

Reviewer #2

(Remarks to the Author)

Wang et al. used TruSeq to increase the quality and annotation of the mRNA transcripts in the sea lamprey retina, thereby enhancing the ability to identify cell types. Subsequently, they characterized the retinal cell types in the lamprey and compared each identified lamprey retinal cell type to jawed vertebrates, revealing conserved and divergent cell types and regulatory networks in evolution. Although it is an interesting study, however, overall, this is a descriptive work of single cell atlas. During the review of this paper, I have the following concerns about this manuscript:

1. Besides providing a single-cell RNA sequencing resource of a new species, does this work uncovered any feature of lamprey retina, either following the trend of evolution or unique in evolution, when compared to other species? Such analysis will push this study to a higher level. Could the authors please address this issue in the manuscript?
2. The authors used several selected species for comparison with the sea lamprey retina in this article. However, it seems that they randomly compared identified lamprey retina cells to equivalent cell types from other species. Why selectively compared BC, SAC, AC, RGC and PR with mice, AC and HC with chickens (Fig 2-6) ? Why not compare all identified cell types with multi-species, e.g. mice and chickens. In addition, MG was not used for comparison. Is MG a unique cell type in lamprey retina? If not, why it was excluded from species comparison?
3. When analyzed TF network in identified cell types from lamprey, macaques were used for comparison (Fig 7)? It is better to compare with all other species, chicken, mice and macaques. It will interpret better the molecular evolution process of the retina.
4. What are the criteria for sub-clustering of each cell type? RGC is a cell type of terminal differentiated cells, and there are many identified subtypes (Fig 6, Fig S2). What is the biological significance? I do believe RGC is also very divergent in the retina from other species. Dose any RGC subtype possibly related to (adapted to) lamprey visual environment, such as water (lamprey) vs land (chicken, mice and macaques); dark vs. light?
5. Related the point 4 listed above, the 4 subtypes of MG are quite far apart in the UMAP plot (Fig 1), but why do they look almost identical (all close to value 1) from Pearson correlation on the hierarchical clustering and heatmap (Fig S2)? PR is the similar situation. Is it due to sequencing quality or some other reason? Please address this issue.
6. Why not use SCENIC for transcription factor analysis? Is the method used to construct the TF network in this article original? Otherwise, please cite the original methodological articles. If not, please include a detailed description in your method section for reproducibility of your analysis. Furthermore, the conservation of transcription factors also needs to be compared among the four species to better explain the evolution of the retina, as mentioned previously.

7. The differences in cell types between species need more detailed analysis, such as differentially expressed genes.
8. How would the authors evaluate the performance achieved by the "XGBoost model"? A supplementary figure and a more detailed description in the method section are needed.
9. The training sets chosen for transcriptomic mapping for each cell type are different (Fig 2-6). What are the criteria for selecting the training sets? And why use different training sets for the analysis?
10. Please include the link of the original datasets of mice, chickens, and macaques used in this study. And describe how these public datasets were processed?

In addition, there are many careless errors that need to be corrected.

- (1) in Fig3c, the immunofluorescence image of SLC17A6 is labeled as c, but not b.
- (2) in Fig4h, scale bar is missing.
- (3) in Fig4i, labels are improper. SAC1/4 and SAC2/3, instead of ON and OFF.
- (4) Lines 247-248, "All AC types expressed a GABA transporter VGT3 (SLC6A11) as well as the glycine transporters GlyT1 (SLC6A5) or GlyT2 (SLC6A9) (Fig. 5a)" is a wrong statement. Based on Fig5a, SLC6A5 is barely expressed in AC2-5, and SLC6A9 is not expressed in AC8. It is very ignorance making such mistake. Please correct.
- (5) in Fig5a, bar indicating Ave.Exp is presented as single value (same color). Similar mistakes apply to entire manuscript. Need to be corrected.
- (6) The description of CAR2 at line 259 was missing in Fig5c.
- (7) In all bubble chart showing gene expression, labels marking gene expression intensity are all missing (e.g. Fig 3h, 4f).
- (8) In Supplementary Fig 5c, it is clear the image is cropped and re-stitched.
- (9) SEG AC and ipRGC appeared only as abbreviation, please provide full name.
- (10) in the legend of Fig7b and 7d, authors wrote "six cell classes were identified". I think you identified 7 cell clusters, right?
- (11) in Fig 7, "g" is missing.

Reviewer #3

(Remarks to the Author)
Review Summary

Wang et al. present an in-depth analysis of the lamprey retina using single transcriptomics, providing a very useful resource for understanding retinal evolution. To do this, they also provide an updated annotation of the current NCBI lamprey genome assembly, which is another very useful resource for the evolutionary biology community. By direct comparison to mouse, chicken and macaque single cell datasets for the retina, they are able to explore the similarities that are conserved across the vertebrates, while also proposing some lamprey specific modifications. Several of their conclusions, however, especially some of those related to proposed lamprey-specific features, require more analysis and support, or are not actually supported by the data that they show. There are also key details that are missing from their main text and the methods section that need to be provided in order for the analyses to be correctly understood and reproduced. I have provided specific details in my comments below, indicating where in the manuscript these changes, corrections or additional analysis are required for publication.

Reviewer comments:

Line 105: Please provide statistics to support this statement.

Line 107: How many genes showed an improvement/change in their gene-body definitions?

Line 117: How many cells did you obtain per replicate?

Line 128: Please provide details of the hierarchical clustering methodology in the methods section or is the phylogenetic tree section of the methods? If so, please make this clear by using consistent terminology.

Line 141: add the references for the mouse and chicken datasets here.

Line 160: By "transcriptomic mapping" are you referring to the XGBoost mapping step in the methods? Please make this clear by using consistent terminology.

Line 176-178: It would be useful to use your analysis of the mouse scRNA-seq data to show that the equivalent cell type in mice does not express SLC17A6 to support this statement.

Line 189 and Line 206: Can you use your data to propose gene markers that distinguish BC7 from the other clusters, since you propose BC7 may be a third, hyperpolarizing type of rod bipolar cell?

Line 209 and Fig 4b: are the markers shown in this figure previously known markers or markers that your analysis identified, or both? Please make this clear.

Line 247-248: Please double check protein and gene names and make sure you are analysing the correct markers.

VGT3= SLC17A8

GAT3 = SLC6A11

Which of these markers do you mean to analyse?

GLYT2= SLC6A5

GLYT1= SCL6A9

Line 247-250 and Fig. 5a: SLC6A5 is not expressed across all the AC groups, suggesting there may be distinct subgroups. Your reference 58 (Yan et al 2020) suggests that Gad1 and Gad2 expression profiles most strongly discriminate between AC subgroups. What do these markers look like in your data? This analysis is needed to support your conclusions.

Line 259: CAR2 is not shown in Fig. 5c. Should this be CA2?

Line 261-262: This conclusion is based on the lack of expression of a single gene. Did you assess the expression of any other gap-junction genes? There are several listed in your updated transcriptome. More evidence is required to support this negative conclusion.

Line 263: AC11 does show some similarity to chicken AC-28 and AC-55, at similar levels to AC10 in the mouse comparison, so this description of your data is not accurate. Please revise.

Line 270: A subset of the RGCs in this group do not seem to express LAMB2 according to Supp. Fig. 6. so I do not think this is good choice of marker for this group if you wish it to encompass the entire group demarcated in orange.

Line 273-274: Are you referring to RGC13 and RGC5 here? Please make this clear. Please also elaborate on what defines and ipRGC.

Line 288: This statement contradicts your statement in Line 273-274 about ipRGCs. Also, it is not clear how you came to this conclusion. There are no ipRGCs listed in the comparison analysis in Fig. 6b and you have not shown a comparison between lamprey and chicken RGCs. This conclusion needs to be backed up with stronger evidence.

Line 300: Please provide a reference to the relevant macaque dataset here and in the methods section. It is also not clear why you chose to move to the macaque here after focussing on the mouse and chicken in other analyses. Please provide an explanation for this choice.

Line 312: It is not clear how these clusters were annotated. Was it based on marker analysis or on the protein activity results? Please make this clear.

Line 386-387: As pointed out above, this conclusion is based on the lack of expression of a single gap junction gene. A more thorough analysis of gap junction gene expression is required to support this conclusion.

Line 393: Please provide a reference for the statement "rods evolved from cones"

Lines 423-428: A search of your Supplementary Table 1 shows that GRM6, TRPM1 and GRIK1 are not present in your transcriptome annotation so could not have shown up in your analysis. It is possible that the markers you found (GRM8, TRPM3 and GRIK2) are actually the closest orthologues of these genes in lamprey. A more thorough phylogenetic analysis of these genes is needed to support your conclusion.

Line 457: Please refer to my comment about Line 247-250. A more thorough analysis of the GABAergic and glycinergic markers is required to support his conclusion.

Line 483-484: You have not presented any data to support his conclusion. This cell type is not referred to in the lamprey-macaque comparison.

Figure and figure legend notes:

Figure 3: Panel label "b" is used twice. One of them should be "c"

Figure 5: Panel label "c" is used twice. One of them should be "d"

Figure 5a: In the legend, the scale bar for expression is lacking a gradient.

Supp. Fig. 5b: In the legend, the scale bar for % match is lacking a gradient.

Fig. 7: label for panel g is missing.

Fig. 7 f: Do the blue dots indicated common master regulators shared with the lamprey? If so, please indicate this in the figure legend.

Line 1229: (e) should be (c)

General notes:

Please italicise all gene names in the text and figures.

Methods:

I see all scripts will be available on GitHub. To allow analysis to be easily repeated, please refer to specific script file names at the appropriate location in the methods section so it is clear which script was used for each part of your analysis.

Line 880: Please provide primer and probe template sequences

Line 878: Please add the reference number for Peng 2019.

Line 883: How long was the Proteinase K treatment done for?

Line 884-885: Was Peng 2019 also followed for the in situ hybridisation procedure? If so, please add the reference here too or if not, give the appropriate reference.

Line 906: how was the RNA obtained? Please provide tissue dissociation and RNA extraction details.

Line 914: spicing should be splicing

Line 924: Please provide details and scripts explaining how this annotation was done

Line 932: Please provide details of the retina dissociation protocol and details of the version of the 10x Genomics kit used. How many cells were targeted and how many cycles were used during the amplification steps?

Line 950 and line 979: What were the parameters for removing low quality cells?

Line 983: Please provide the details of the mouse and chicken datasets used, including links to the appropriate data repositories.

Line 996 and line 1076: What were the downsampling parameters?

Version 1:

Reviewer comments:

Reviewer #2

(Remarks to the Author)

Dear Editor,

the authors have very well addressed all my concerns. Thank you! I am satisfied with the revisions and think it is now ready for publishing.

(Remarks on code availability)

Reviewer #3

(Remarks to the Author)

The responses to my comments, new analyses and updates to the manuscript that the authors have provided have dealt with the majority of my concerns. I am grateful for their extra work.

I have some minor concerns about how they have addressed some of my comments and a few new, minor suggestions for corrections. I have copied my original comments and their responses below. I have added my new comments related to each point in **bold**.

Line 107: How many genes showed an improvement/change in their gene-body definitions?

After the update, the NCBI+TruSeq reference transcriptome had 15,070 genes updated by StringTie with MSTRG numbers, suggesting changes in their gene body definitions (Supplementary Fig. 1c). A detailed comparison between the NCBI and NCBI+TruSeq references using gffcompare revealed 15,871 novel exons, 11,391 novel introns, and 3,430 novel loci in the NCBI+TruSeq reference transcriptome.

Thank you for this additional gffcompare analysis which addresses my comment. Please include this information in the manuscript. Also, please note, the reference to Supplementary Fig. 1c is no longer in the manuscript. Please add this back.

Line 117: How many cells did you obtain per replicate?

We obtained 12,021 and 9,453 cells from the S1 and S2 replicates, respectively.

Please add this information to line 115.

Line 176-178: It would be useful to use your analysis of the mouse scRNA-seq data to show that the equivalent cell type in mice does not express SLC17A6 to support this statement.

The lack of SLC17A6 expression in BCs, while present in RGCs, has been demonstrated in Figure 5D of Macosko et al., 2015 (Ref 29) and Figure 1C of Yan et al., 2020 (Ref 34). We have included these references in the corresponding statement.

Please add ref 29 to Line 185. It is missing.

Line 247-250 and Fig. 5a: SLC6A5 is not expressed across all the AC groups, suggesting there may be distinct subgroups. Your reference 58 (Yan et al 2020) suggests that Gad1 and Gad2 expression profiles most strongly discriminate between AC subgroups. What do these markers look like in your data? This analysis is needed to support your conclusions.

We apologize for the confusion. We have revised the sentences. GAD2 is not found in the lamprey genome. We have plotted the expression of GAD1, SLC32A1 (VGAT), SLC6A1 (GAT1), SLC6A11 (GAT3), SLC6A9 (GLYT1), and SLC6A5 (GLYT2) in all AC types in Supplementary Fig. 6b. SLC6A13 (GAT2) is not found in the lamprey genome, and the expression of SLC6A1 is very low in all AC types.

All AC types except AC8 express both SLC6A11 (GAT3) and SLC6A9 (GLYT1), suggesting they are both GABAergic and glycinergic. AC8 expresses only SLC6A5, indicating that it is a glycinergic AC type.

In line 279 in the revised manuscript you state: “AC8 is uniquely glycinergic with the expression of only the glycine transporter SLC6A9 (GLYT1)”. Given your statement in response to my comment above and the data in Supplementary Fig. 6, which shows SLC6A9 is not expressed by AC8, this should be SLC6A5 (GLYT2). Please correct this.

Interestingly, among all the GABAergic and glycinergic ACs, AC1, AC6, AC7, AC9, AC10, and AC11 express both glycine transporter SLC6A5 and SLC6A9, while AC2-5 only express one glycine transporter SLC6A9.

In line 280, AC7 is missing from this list. Please add it in.

Line 288: This statement contradicts your statement in Line 273-274 about ipRGCs. Also, it is not clear how you came to this conclusion. There are no ipRGCs listed in the comparison analysis in Fig. 6b and you have not shown a comparison between lamprey and chicken RGCs. This conclusion needs to be backed up with stronger evidence.

We have showed that RGC5 and 13 have the highest correspondence percentage with mouse M1 ipRGCs. We also included the comparison of RGCs between lamprey and chicken and also between lamprey and zebrafish.

Thank you for this clarification. It would be helpful to add the specific names of the M1 ipRGCs that are used in the figures to line 327 so it is clear which cell types you are referring to.

New comments:

- **Line 297-298: The violin plots in Supp. Fig. 6c indicate that some cells in these groups do express these genes. The sentence should rather read: “We detected very little expression of any other lamprey gap-junctional proteins in AC8”**
- **Supplementary Figure 6 legend: Panels “b” and “c” show violin plots, not dot plots. Please correct.**
- **You have not referred to Supplementary figure 9 in the text. Please add it in.**
- **Line 1280: Should this be Supplementary Table 7? There is no Supplementary Table 9.**

(Remarks on code availability)

The link is not working.

RESPONSE TO REVIEWERS' COMMENTS

Reviewer #1 (Remarks to the Author):

These authors have an authoritative perspective on the evolution and physiology of the lamprey retina. They are the experts. Here, they expand this field by re-investigating old & unresolved questions about the agnathan retina cell types. With that new data in hand, new ideas can be presented about the earliest vertebrate retinas. Considering the long history of fascination with eye evolution across centuries, which continues to this day, the current manuscript will be of broad interest. The work is completed with great care and the analysis of cell types and master regulators seems to be sound. Congratulations on another landmark contribution.

Thank you very much. We greatly appreciate your positive evaluation of our work.

The authors have chosen to use mouse as the representative gnathostome for most of their analyses. This is unfortunate because mice (like all mammals) have a retina that has gone through dramatic regressive phases of evolution and no longer has the characters of a typical gnathostome retina. Happily, this is not causing the authors too many issues of interpretation for many of their comparisons, e.g. where the authors are pointing out similarities between cell types in lampreys and gnathostome (e.g. the RBCs, the PR). However it creates a large challenge of interpreting the situation when noting that the cell types do not align well – e.g. The abstract reads “...conserved cell types shared between jawless and jawed lineages [are impressive]... In contrast to this evidence for conservation, the pathways of diversification for amacrine cells and retinal ganglion cells appear to have distinctly diverged between the two lineages.”. A more likely interpretation is that mammals (especially the nocturnal clinically blind mice) have diverged away from other gnathostomes as they regressed into blindness. Two good solutions to this would be to i) alter the wording to not suggest mice are an exemplar of gnathostomes [e.g. in the abstract the authors are comparing lamprey vs. mice, not jawed vs. jawless]; ii) compare the lamprey data to chicken or frogs or fish. I think this paucity of comparison to non-mammalian gnathostomes, if it remains in the manuscript, warrants comment in the section “limitations of our study”.

Thank you for the excellent point. We have reworded the abstract and acknowledged the limitation of cross-species comparisons in the “Limitations of our study” section. Given the shared comments with Reviewer #2, we have also included BCs and RGCs from chicken and zebrafish into the cross-species comparison (Fig. 3h-j, 6; Supplementary Fig. 4, 5, 8). With these additional comparisons beyond the mouse, we still observed similar conservation patterns and divergent paths.

A paper by Hahn et al, cited by the authors as reference 21 and a preprint in BioRxiv, is now published in Nature, <https://doi.org/10.1038/s41586-023-06638-9>, and largely contains similar conclusions to the current submission but with 17 species compared rather than a focus on the lamprey. I welcome the focus on lamprey as a window into appreciating the evolutionary origins of the vertebrate retina, and so I think the current contribution will be impactful on the field. Some text, e.g. line 66, could be rephrased to reflect this recent work.

We highly appreciate your recognition of the importance of studying the lamprey retina. We have revised and rephrased the relevant text to reflect the recent publication.

The authors provide useful validation of their RNA-Seq data by including in situ hybridizations. Some further details would make the latter a lot more convincing: a Supplemental Table with details about the riboprobes used (primers, probe length). The authors do a very good job of convincing the riboprobes are not suffering technical artefacts (e.g. the anti-DIG-HRP and Anti-Fir-POD are actually being very well differentiated, despite both being peroxidase), but they do not really communicate this in their Methods or Results – Supplemental Fig 3 shows great experimental care.

We have included **Supplementary Table 3**, which provides detailed information about all the RNA probes used in this study.

Double fluorescence *in situ* uses a similar peroxidase reaction to develop two fluorescent colors. However, the distinction between these two colors relies on a sequential development process. Briefly, after hybridizing both probes, only one probe will be immunostained first with its corresponding HRP- or POD-conjugated antibody. After the development of the first probe's fluorescent color, the HRP or POD activity is quenched with 3% H₂O₂ before incubating the second probe with its corresponding HRP- or POD-conjugated antibody. The **Methods "Fluorescence *In Situ* Hybridization Validations"** section now has included more details on double fluorescence *in situ*.

The manuscript would be strengthened by detailing if the key gene comparisons are orthologous. E.g. a synteny analysis for key genes that discriminate cell types would be appropriate.

Thank you for the good suggestion. We have included the detailed orthologous relationships among lamprey, zebrafish, chicken, mouse, macaque, and human genes in **Supplementary Table 4**.

Minor ideas: (I read the title & abstract and then looked at the Figs, as is typical, so some comments may be moot once the reader digs into the main text).

a) Fig 1b the schematics are intuitive to me as a retina aficionado, but I wonder if there is space in the legend to define the grey blobs (nuclei).

Thank you for the suggestion. We have revised the diagram of retinal structures for both cyclostome and gnathostome retina and added definition of the nuclei position, which is marked by grey boxes in the legend.

b) Fig 1e suggest labelling the ONL, INL, etc.. I'm not 100% sure what I am looking at... Same for Fig S3 (looking at S3 makes me even less sure about what layers Fig 1e is showing, but with some labelling of layers I think you could have it accomplish the opposite).

We apologize for the confusion. To aid in visualization, we have demarcated the retinal layers in **Fig. 1e, 2c, 3c, 3d, and Supplementary Fig. 3a, and 3b**.

c) Figure 3: panel c (FISH) is accidentally labelled with a "b"

We have corrected it.

d) The order of ideas in line 53-57 could be more pleasing – rods vs. cones is spread out among intervening thoughts.

We have changed the order of ideas placing the rods vs. cones comment first.

e) Where is the RPE? And I'm always confused by retina scRNA-seq papers – aren't there any blood cells or microglia? If these were ignored, then that is very reasonable but could be stated in the Methods or perhaps at the start of the Results (that begins with an overview of the authors' approach)

The lack of RPE cells in our study and other scRNA-seq studies of the retina is due to tissue preparation. We isolated the retina from the choroid, which separated the RPE layer from the retina. Even if some RPE cells remained attached to the retina, their number was too sparse to be detected by scRNA-seq. Studies utilizing RPE cells typically involve the dissociation of the choroid tissue (for example: PMID: 31712411). We have added this information in the **Methods "Single Cell RNA-sequencing Library Preparation."**

Blood cells are rarely found in scRNA-seq analyses of the retinal tissue, likely due to being lost during washing steps. In studies of primate retinas (for example, PMID: 3859343, PMID: 30712875, and PMID: 3255229), pericytes and endothelial cells from the blood vessel can be detected, and occasionally microglia. However, their fractions are extremely low. In this study of lamprey, we didn't detect these cell types.

As a result, we didn't detect RPE, microglia, blood cells, or other immune cells in our dataset. There was no expression of marker genes for RPE, microglia, or other non-neuronal cell types. Please see the **Supplementary Fig. 1d**. We have added one sentence in the results to acknowledge this observation.

f) Similarly, the interpretation is centered on differentiated cell types – were cells in earlier stage of development excluded?

We used post-metamorphic, sexually matured, adult lamprey for this study. Unlike teleost and amphibian retinas, there are no proliferating cells observed in the lamprey retina after metamorphosis (PMID: 18295752). Therefore, we didn't filter out any proliferating cells, and all the cells that we analyzed in the adult stage are fully differentiated.

g) The first section of the Methods "Tissue Procurement" would benefit from some additional details. E.g. these were post-metamorphic lamprey? Sexually mature? What does "posterior eyecup" include [sclera, RPE, choroid]?

As explained above, the lamprey used in this study were post-metamorphic and sexually mature. We added this information to the **Methods "Tissue Procurement."**

h) The writing on line 360 could be reworked to include the concept of regression. Much of the differences between lamprey and mouse (or lamprey and macaque) must be driven by the nocturnal bottleneck and loss of retinal complexity at the emergence of the mammals. I'm only encouraging a brief acknowledgment, e.g. some cells were lost while others were preserved....

A good point. We now say, "Our work suggests that evolution proceeded opportunistically, preserving cells and circuits that maintained their usefulness, losing other cells and circuits that were no longer relevant, and inventing new mechanisms as these became adaptive."

i) Line 365 is a bit overstated (the whole section is a bit discombobulated but the connections of ideas are appreciated). Your data speak to cell types being present, and so you can say that the cell substrates that underpin rod circuitry are there, but you cannot say much yet about circuitry. You've also almost ignored cone circuitry in this section (which is fine but then it is not about duplex retina), and the writing is framed in a way that fails to acknowledge that mammal rod circuitry is not a good representation of gnathostomes (so it cannot be used to infer the origins!). Kudos for corrective writing that acknowledges some of the complexity beginning on line 391.

We agree that mammalian rod circuitry is not a good representation of gnathostomes and have revised the sentence to minimize the confusion. We develop these ideas in much greater detail in a manuscript in progress.

j) Line 452 typo? Mouse has two (smudgy and regressed) cone types.

We have revised the paragraph.

k) Line 448 I agree that retaining four HC types in lamprey is fascinating. How does one decide that there is only one type of cone in lamprey? E.g. there must be two or three subtypes within the cones you detected (even if they all share only one opsin type)? No need to address this silly point, but I don't really understand where one decides to be a lumpner or splitter among these (or any) cell types – surely not all the cones you detected were identical....

The evidence for only a single cone type in *Petromyzon marinus* is morphological from an old but fairly solid literature, and from microspectrophotometry from our own work and from the earlier work of Hárosi and Kleinschmidt, cited in our paper. However, we agree that some other lamprey species, such

as the southern hemisphere pouched lamprey, have five different photoreceptor types, each expressing one of the five different visual opsin gene classes.

l) Line 457 again you assume that mouse retina is more advanced than lamprey retina. Evolution is not always so progressive, and mice have lost lots compared to fish and birds. But you are just speculating about the history here, so I guess it is ok.

As you say, history, “wie es eigentlich gewesen . . .”

Reviewer #2 (Remarks to the Author):

Wang et al. used TruSeq to increase the quality and annotation of the mRNA transcripts in the sea lamprey retina, thereby enhancing the ability to identify cell types. Subsequently, they characterized the retinal cell types in the lamprey and compared each identified lamprey retinal cell type to jawed vertebrates, revealing conserved and divergent cell types and regulatory networks in evolution. Although it is an interesting study, however, overall, this is a descriptive work of single cell atlas. During the review of this paper, I have the following concerns about this manuscript:

1. Besides providing a single-cell RNA sequencing resource of a new species, does this work uncovered any feature of lamprey retina, either following the trend of evolution or unique in evolution, when compared to other species? Such analysis will push this study to a higher level. Could the authors please address this issue in the manuscript?

Thank you for the good suggestion. We apologize for not clearly highlighting the unique features of the lamprey retina. We have added new analysis to further emphasize these characteristics.

Our study not only provides a cell atlas of the lamprey retina but also uncovers specific features of the lamprey retina. The most distinct feature of the lamprey retina is the differentiation of its inhibitory neuronal class, amacrine cells (AC). First, the lamprey has fewer AC type types, and almost all types appear to be dual-transmitter neurons (GABAergic and glycinergic), unlike jawed vertebrates where most AC types are typically either GABAergic or glycinergic but not both (**Supplementary Fig. 6a, b**). Second, lamprey ACs have fewer conserved genetic regulators compared to jawed species (**Fig. 7d, e**). Thus, lamprey ACs haven't acquired functional similarity to those in jawed species, suggesting a major divergence between the two lineages. For instance, All-like amacrine cells in lamprey seem to lack gap-junction channels, so the rod pathway circuit might differ from that in jawed species. Interestingly, we show that each lamprey amacrine cell type is represented by multiple types in mammals (**Fig. 5c, d**), suggesting extensive radiation of amacrine cells during vertebrate evolution and evolutionary pressure on diversifying AC types in jawed species.

Another divergent feature can be seen from our analysis of lamprey RGCs. Despite a similar number of RGC types between the lamprey and jawed species, many lamprey types don't correspond to any type in comparison to jawed species. This finding could reflect the unique somatic position and axonal projections of lamprey ganglion cells, indicating different roles for these cells in animals living in different environments with varying visual demands (**Fig. 1a, 6d-f**).

Lastly, our analysis shows that lamprey have both ON and OFF rod bipolar cells (**Fig. 3b**), unlike mammals, which only have ON rod bipolar cells. Since amphibians also have rod ON and OFF bipolar cells, our analysis indicates that ON and OFF rod bipolar cells likely represent the primitive state, and that OFF rod bipolar cells were lost (and replaced by All amacrine-cell circuits) at some point between amphibians and mammals.

In contrast to these unique features of lamprey retinal cells, homologous cell types between the lamprey and jawed species empathize a striking conservation of the vertebrate retina, extending from the previously known six cell classes to multiple terminal cell types. This conservation illuminates the evolution and Cambrian origin of cell types in the vertebrate retina, which has been more difficult to examine in other parts of the central nervous system.

In addition to these concepts, our methods for identifying homologous cell types across a long phylogenetic tree will be valuable for the study of similar questions in other biological systems. We have strengthened the discussion of these points in the results and discussion.

2. The authors used several selected species for comparison with the sea lamprey retina in this article. However, it seems that they randomly compared identified lamprey retina cells to equivalent cell types from other species. Why selectively compared BC, SAC, AC, RGC and PR with mice, AC and HC with chickens (Fig 2-6)? Why not compare all identified cell types with multi-species, e.g. mice and chickens. In addition, MG was not used for comparison. Is MG a unique cell type in lamprey retina? If not, why it was excluded from species comparison?

We have extended the species comparisons from mouse to chicken and, in some cases, also zebrafish (Fig. 3, 6, Supplementary Fig. 4, 5, 8). The primary reasons for species selection were the availability of scRNA-seq datasets and comprehensive cell-type characterization (Supplementary Table 2). For example, zebrafish AC types haven't been well characterized by scRNA-seq, leading to their exclusion from the AC comparison. Another consideration was the diversity of cell types for comparison in particular species. For example, as mice possess only one HC type, we used chicken HCs for our comparison. Chicken exhibit a similar number (4) of HC types as lamprey, allowing us to avoid misrepresenting lamprey-specific features. We have added additional explanations to the corresponding text.

MG is not a unique type in the lamprey. We excluded it from cell type comparison as its gene expression can be rapidly dynamic depending on developmental stages and its localization in the retina. Therefore, we focused on cell types in neuronal cell classes, such as PR, HC, BC, AC, and RGC for comparisons. However, we did compare the genetic programs specifying the MG class between lamprey and jawed species (Fig. 7d, e, Supplementary Fig. 10, 11).

3. When analyzed TF network in identified cell types from lamprey, macaques were used for comparison (Fig 7)? It is better to compare with all other species, chicken, mice and macaques. It will interpret better the molecular evolution process of the retina.

We have now included mouse and chicken in the analysis (Fig. 7, Supplementary Fig. 10, 11).

4. What are the criteria for sub-clustering of each cell type? RGC is a cell type of terminal differentiated cells, and there are many identified subtypes (Fig 6, Fig S2). What is the biological significance? I do believe RGC is also very divergent in the retina from other species. Dose any RGC subtype possibly related to (adapted to) lamprey visual environment, such as water (lamprey) vs land (chicken, mice and macaques); dark vs. light?

In the retina, there are different levels of heterogeneity among cell types within each cell class. For example, RGCs, ACs, and BCs have a higher diversity of cell types compared to other classes. To better characterize all types within each class, it is common practice to subset each cell class and analyze cell types within each subset separately (such as, PMID: 37388908, Refs 19, 21, 30). The criteria for sub-clustering of the cell type have been described in the **Methods "Cell-type classification within each class" section**.

As the sole output neurons of the retina, RGCs can be classified into over 40 types, each computing specific visual features such as luminance, motion, direction, and color (Ref 80). Collectively, all RGC types transmit this information to the rest of brain to form an image. Thus, the diversity and function of RGC types reflect how visual information is encoded in each species. By comparing lamprey RGC types with those of mouse, chicken, and zebrafish (Fig. 6), our data suggested that lamprey might possess RGC types analogous to direction-selective ooDSGCs in mice, motion-detection W3B in mice, and ipRGCs involved in circadian entrainment and pupillary response. We have elaborated on this in the corresponding results and discussion sections. We also identified potential lamprey-specific RGC types lacking counterparts in jawed species. These non-conserved RGC types might have distinct

somatic positions and axonal projections (**Fig. 6d-f**). However, we now acknowledge the limitation of lacking direct functional measurements for these RGCs.

5. Related to the point 4 listed above, the 4 subtypes of MG are quite far apart in the UMAP plot (Fig 1), but why do they look almost identical (all close to value 1) from Pearson correlation on the hierarchical clustering and heatmap (Fig S2)? PR is the similar situation. Is it due to sequencing quality or some other reason? Please address this issue.

In Figure 1F, the clusters of MG types in the UMAP are constructed from highly variable genes among all MG, aiming to identify all heterogeneous types within the MG class. In Supplementary Fig. 2, Pearson correlations are calculated based on highly variable genes among all six cell classes (PR, HC, BC, AC, RGC, and MG), generally emphasizing the correlation of cell types within individual cell classes. As distinct sets of highly variable genes are used for UMAP clustering and hierarchical clustering, the results exhibit different correlations and are not comparable. We have elaborated on this point in the corresponding method sections.

6. Why not use SCENIC for transcription factor analysis? Is the method used to construct the TF network in this article original? Otherwise, please cite the original methodological articles. If not, please include a detailed description in your method section for reproducibility of your analysis. Furthermore, the conservation of transcription factors also needs to be compared among the four species to better explain the evolution of the retina, as mentioned previously.

There are several limitations for the application of SCENIC in non-model organisms. First, the SCENIC workflow relies on species-specific cisTarget databases to refine candidate co-expression gene modules to mature regulons. Currently, pre-built databases are available for a limited set of model species, including human, fruit fly, and mouse (<https://resources.aertslab.org/cistarget/databases/>). For other species, constructing a species-specific cisTarget database is necessary, which involves scoring each TF motif against the regulatory regions of each gene. This presents a challenge for the lamprey, as its genome lacks well-annotated regulatory regions. In addition, the SCENIC pipeline is not designed to infer the regulatory activity of surface proteins or signaling molecules, which may also play critical roles in cell function.

To overcome these limitations, we used a published method, ARACNe-AP (Ref 87), to reverse engineer regulatory networks of regulator genes from scRNA-seq datasets. We subsequently developed a new algorithm, ROSA, to quantify the protein activities of these regulators using a statistical model. This novel approach enables the inference of activities for various candidate regulators, including TFs, coTFs, membrane proteins, and signaling molecules across multiple species. A detailed description of the ROSA algorithm and protein activity inference has been provided in the **Methods “Inferring Protein Activity of Genetic Regulators”** section. Leveraging this analytical framework, we have compared the conservation of gene regulators among all four species, extending our analysis beyond the lamprey-macaque comparison (**Fig. 7, Supplementary Fig. 10, 11**).

7. The differences in cell types between species need more detailed analysis, such as differentially expressed genes.

Thank you for the suggestion. We have added differentially expressed genes between species for RGCs and BCs in **Fig. 6d-f**, and **Supplementary Fig. 5g-i**.

8. How would the authors evaluate the performance achieved by the “XGBoost model”? A supplementary figure and a more detailed description in the method section are needed.

We have illustrated the cross-species integration and XGBoost mapping in **Supplementary Fig. 4c** (iv) and provided detailed methods in **“Transcriptomic mapping via XGBoost.”** Briefly, to construct and evaluate the XGBoost classification model, we initially split the reference dataset (cell types in one species) into a training dataset (67% of full data) and a validation dataset (33%). A classification model predicting targeted cell-type labels was built from the training dataset, with its performance assessed on the validation set. Following successful model validation with high

confidence predictions, we applied the actual testing dataset (cell types from another species) to the model and generated correspondence predictions, visualized in a confusion matrix. In addition, we computed normalized entropy scores for these predictions. A lower normalized entropy score signifies greater prediction confidence. See **Supplementary Fig. 8h**, and the **Methods “Integration Performance Comparison using Entropy Analysis.”**

9. The training sets chosen for transcriptomic mapping for each cell type are different (Fig 2-6). What are the criteria for selecting the training sets? And why use different training sets for the analysis?

For Fig. 2, 3, and 4, since cell types in jawed species are well characterized, we used them as a training set/reference to annotate the function of lamprey cell types. For Fig. 5 and 6, our goal was to determine the extent to which any type in each of these selected jawed species could be correlated to lamprey AC and RGC types. We thus used lamprey cell types as the training set, enabling the identification of lamprey-specific cell types.

10. Please include the link of the original datasets of mice, chickens, and macaques used in this study. And describe how these public datasets were processed?

We have summarized the original datasets of mice, chicken, zebrafish and macaque in **Supplementary Table 2**. If the processed data is used, we have provided links to the processed dataset. For the raw datasets, we preprocessed the data following the Seurat guided clustering tutorial (https://satijalab.org/seurat/articles/pbmc3k_tutorial).

In addition, there are many careless errors that need to be corrected.

We apologize for all these errors and have attempted to correct them.

(1) in Fig3c, the immunofluorescence image of SLC17A6 is labeled as c, but not b.

We have corrected this mistake.

(2) in Fig4h, scale bar is missing.

In Fig.4h, scale bars were only placed in the merged images (DAPI + *CHAT* or DAPI + *MEGF10*). We omitted the scale bar in the individual grayscale images (*CHAT* or *MEGF10* alone) as they were the same as showed in the merged images. We followed this rule consistently throughout the manuscript.

(3) in Fig4i, labels are improper. SAC1/4 and SAC2/3, instead of ON and OFF.

We have corrected the labels in Fig. 4i.

(4) Lines 247-248, “All AC types expressed a GABA transporter VGT3 (SLC6A11) as well as the glycine transporters GlyT1 (SLC6A5) or GlyT2 (SLC6A9) (Fig. 5a)” is a wrong statement. Based on Fig5a, SLC6A5 is barely expressed in AC2-5, and SLC6A9 is not expressed in AC8. It is very ignorance making such mistake. Please correct.

We apologize for any confusion and have rewritten this section to improve the clarity. While *SLC6A5* is absent from AC2-5 and *SLC6A9* is absent from AC8, all AC types express either *SLC6A5* alone, *SLC6A9* alone, or both. This means that they are all potentially glycinergic. In addition, all ACs express the GABA transporter *SLC6A11*, albeit at low levels. The combined expression of GABA and glycine transporters suggests that nearly all AC types are both GABAergic and glycinergic, though they may use different transporters for glycine.

(5) in Fig5a, bar indicating Ave.Exp is presented as single value (same color). Similar mistakes apply to entire manuscript. Need to be corrected.

We have corrected all the bars throughout the manuscript.

(6) The description of CAR2 at line 259 was missing in Fig5c.

CAR2 is the same gene as CA2. We added this explanation in the text, figure, and figure legend.

(7) In all bubble chart showing gene expression, labels marking gene expression intensity are all missing (e.g. Fig 3h, 4f).

We have added the labels indicating gene-expression intensity.

(8) in Supplementary Fig 5c, it is clear the image is cropped and re-stitched.

We have corrected Supplementary Fig. 5c, which is now Fig. 5d

(9) SEG AC and ipRGC appeared only as abbreviation, please provide full name.

We have provided their full names in either the main text or the corresponding figure legend.

(10) in the legend of Fig7b and 7d, authors wrote “six cell classes were identified”. I think you identified 7 cell clusters, right?

As SACs are dominant amacrine cells in the lamprey, we divided the AC class into SAC and non-SAC subclasses. In the revised Fig. 7, we group them together as one AC class (**Fig. 7b**).

(11) in Fig 7, "g" is missing.

Fig. 7 has been updated.

Reviewer #3 (Remarks to the Author):

Review Summary

Wang et al. present an in-depth analysis of the lamprey retina using single transcriptomics, providing a very useful resource for understanding retinal evolution. To do this, they also provide an updated annotation of the current NCBI lamprey genome assembly, which is another very useful resource for the evolutionary biology community. By direct comparison to mouse, chicken and macaque single cell datasets for the retina, they are able to explore the similarities that are conserved across the vertebrates, while also proposing some lamprey specific modifications. Several of their conclusions, however, especially some of those related to proposed lamprey-specific features, require more analysis and support, or are not actually supported by the data that they show. There are also key details that are missing from their main text and the methods section that need to be provided in order for the analyses to be correctly understood and reproduced. I have provided specific details in my comments below, indicating where in the manuscript these changes, corrections or additional analysis are required for publication.

Thank you for your positive assessment of our research. We appreciate your detailed comments, which significantly helped us improve the quality of our manuscript.

Reviewer comments:

Line 105: Please provide statistics to support this statement.

Since there are only two replicates used for the alignment, we can't conduct a statistical analysis. Thus, we have removed the term “significantly” from this sentence.

Line 107: How many genes showed an improvement/change in their gene-body definitions?

After the update, the NCBI+TruSeq reference transcriptome had 15,070 genes updated by StringTie with MSTRG numbers, suggesting changes in their gene body definitions (**Supplementary Fig. 1c**). A detailed comparison between the NCBI and NCBI+TruSeq references using gffcompare revealed 15,871 novel exons, 11,391 novel introns, and 3,430 novel loci in the NCBI+TruSeq reference transcriptome.

Line 117: How many cells did you obtain per replicate?

We obtained 12,021 and 9,453 cells from the S1 and S2 replicates, respectively.

Line 128: Please provide details of the hierarchical clustering methodology in the methods section or in the phylogenetic tree section of the methods? If so, please make this clear by using consistent terminology.

Thank you for the suggestion. The hierarchical clustering methodology has been described in the **Methods “Hierarchical Clustering Analysis”** section. To ensure consistency and clarity, we have revised this section by adding more details.

Line 141: add the references for the mouse and chicken datasets here.

We have added references and included this information in **Supplementary Table 2**.

Line 160: By “transcriptomic mapping” are you referring to the XGBoost mapping step in the methods? Please make this clear by using consistent terminology.

Yes, the transcriptomic mapping was achieved with XGBoost (eXtreme Gradient Boosting). To be consistent, we have changed “transcriptomics mapping” to “transcriptomics mapping via XGBoost” throughout the manuscript.

Line 176-178: It would be useful to use your analysis of the mouse scRNA-seq data to show that the equivalent cell type in mice does not express SLC17A6 to support this statement.

The lack of *SLC17A6* expression in BCs, while present in RGCs, has been demonstrated in Figure 5D of Macosko et al., 2015 (Ref 29) and Figure 1C of Yan et al., 2020 (Ref 34). We have included these references in the corresponding statement.

Line 189 and Line 206: Can you use your data to propose gene markers that distinguish BC7 from the other clusters, since you propose BC7 may be a third, hyperpolarizing type of rod bipolar cell?

These gene markers for BC7 are shown in **Fig. 3b**.

Line 209 and Fig 4b: are the markers shown in this figure previously known markers or markers that your analysis identified, or both? Please make this clear.

The melanopsin-like gene was shown to be expressed in the horizontal cells of the river lamprey (Ref 63). Our scRNA-seq and fluorescence in situ results have confirmed a similar expression pattern in sea lamprey HCs. We have reworded the sentence to make prior knowledge and our findings clearly distinguishable.

Line 247-248: Please double check protein and gene names and make sure you are analysing the correct markers.

VGT3= SLC17A8
GAT3 = SLC6A11

Which of these markers do you mean to analyse?

GLYT2= SLC6A5

GLYT1= SCL6A9

We sincerely apologize for these mistakes. We have double checked the protein and gene names and corrected these errors. GAT3 = SLC6A11, GLYT2= SLC6A5, GLYT1= SLC6A9.

Line 247-250 and Fig. 5a: SLC6A5 is not expressed across all the AC groups, suggesting there may be distinct subgroups. Your reference 58 (Yan et al 2020) suggests that Gad1 and Gad2 expression profiles most strongly discriminate between AC subgroups. What do these markers look like in your data? This analysis is needed to support your conclusions.

We apologize for the confusion. We have revised the sentences. GAD2 is not found in the lamprey genome. We have plotted the expression of GAD1, SLC32A1 (VGAT), SLC6A1 (GAT1), SLC6A11 (GAT3), SLC6A9 (GLYT1), and SLC6A5 (GLYT2) in all AC types in **Supplementary Fig. 6b**. SLC6A13 (GAT2) is not found in the lamprey genome, and the expression of SLC6A1 is very low in all AC types.

All AC types except AC8 expresses both SLC6A11 (GAT3) and SLC6A9 (GLYT1), suggesting they are both GABAergic and glycinergic. AC8 expresses only SLC6A5, indicating that it is a glycinergic AC type. Interestingly, among all the GABAergic and glycinergic ACs, AC1, AC6, AC7, AC9, AC10, and AC11 express both glycine transporter SLC6A5 and SLC6A9, while AC2-5 only express one glycine transporter SLC6A9. This finding suggests that not all AC types may be clearly separated into GABAergic or glycinergic classes but could nevertheless be distinguished by the usage of different glycinergic transporters.

Line 259: CAR2 is not shown in Fig. 5c. Should this be CA2?

Yes, CA2 is also known as CAR2. We have added this information in the text and figure legend of Fig. 5c.

Line 261-262: This conclusion is based on the lack of expression of a single gene. Did you assess the expression of any other gap-junction genes? There are several listed in your updated transcriptome. More evidence is required to support this negative conclusion.

Yes, we have checked all the gap-junction genes that can be found in the lamprey genome in **Supplementary Fig. 6c**. Only GJD2 is expressed and only by AC5. We have clarified the conclusion.

Line 263: AC11 does show some similarity to chicken AC-28 and AC-55, at similar levels to AC10 in the mouse comparison, so this description of your data is not accurate. Please revise.

Thank you for the observation. We have revised the conclusion.

Line 270: A subset of the RGCs in this group do not seem to express LAMB2 according to Supp. Fig. 6. so I do not think this is good choice of marker for this group if you wish it to encompass the entire group demarcated in orange.

Thank you for the suggestion. We have replaced it with another marker, LRFN4.

Line 273-274: Are you referring to RGC13 and RGC5 here? Please make this clear. Please also elaborate on what defines and ipRGC.

Yes, RGC5 and RGC13 are the two ipRGC types in the lamprey. We have elaborated on the definition of ipRGCs and why we identified RGC5 and 13 as ipRGCs in the main text.

Line 288: This statement contradicts your statement in Line 273-274 about ipRGCs. Also, it is not clear

how you came to this conclusion. There are no ipRGCs listed in the comparison analysis in Fig. 6b and you have not shown a comparison between lamprey and chicken RGCs. This conclusion needs to be backed up with stronger evidence.

We have showed that RGC5 and 13 have the highest correspondence percentage with mouse M1 ipRGCs. We also included the comparison of RGCs between lamprey and chicken and also between lamprey and zebrafish.

Line 300: Please provide a reference to the relevant macaque dataset here and in the methods section. Its is also not clear why you chose to move to the macaque here after focusing on the mouse and chicken in other analyses. Please provide an explanation for this choice.

Thank you for the suggestion. We have included mouse and chick for the comparison in Fig. 7.

Line 312: It is not clear how these clusters were annotated. Was it based on marker analysis or on the protein activity results? Please make this clear.

Dimension-reduction analysis and construction of the KNN graph for clustering were performed using the protein activities of inferred regulators. Following clustering, cells were annotated based on cell-class identities determined by scRNA-seq analysis (marker analysis) in Fig. 1c. Our results showed that the clusters based on protein activities of regulators (Fig. 7b) closely align with the cell classes identified based on gene expression. We have clarified this in the corresponding figure legend.

Line 386-387: As pointed out above, this conclusion is based on the lack of expression of a single gap junction gene. A more thorough analysis of gap junction gene expression is required to support this conclusion.

See our statement above for lines 261 – 262.

Line 393: Please provide a reference for the statement “rods evolved from cones”

The best evidence for the later emergence of rods can be found from an analysis of rod and cone visual pigments, as in Shichida Y, Matsuyama T (2009) Evolution of opsins and phototransduction. *Philos Trans R Soc Lond B Biol Sci* 364:2881-2895.

Lines 423-428: A search of your Supplementary Table 1 shows that GRM6, TRPM1 and GRIK1 are not present in your transcriptome annotation so could not have shown up in your analysis. It is possible that the markers you found (GRM8, TRPM3 and GRIK2) are actually the closest orthologues of these genes in lamprey. A more thorough phylogenetic analysis of these genes is needed to support your conclusion.

Thank you for the insightful suggestion. We performed phylogenetic analysis of these genes among lamprey, zebrafish, chicken, mouse, macaque, and human using OrthoFinder (**Supplementary Table 4, Supplementary Fig. 4c-f**). These analyses showed that lamprey *GRIK2*, *GRM8.2*, and *TRPM3.1* are the closest orthologous genes to *Grik1*, *Grm6*, and *Trpm1* in jawed species. Thus, the gene usage among ON and OFF BC types in the lamprey is similar to that in jawed species. We have revised this section to reflect these new analyses.

Line 457: Please refer to my comment about Line 247-250. A more thorough analysis of the GABAergic and glycinergic markers is required to support his conclusion.

We have added **Supplementary Fig. 6b** to show the expression of all the GABAergic and glycinergic markers among all AC types. We have revised the conclusion to make it clearer and more accurate.

Line 483-484: You have not presented any data to support his conclusion. This cell type is not referred to in the lamprey-macaque comparison.

We found that lamprey ipRGCs (RGC5 and RGC13) showed a high correspondence with mouse M1 ipRGCs (**Fig. 6b, c**). We have revised the conclusion.

Figure and figure legend notes:

Figure 3: Panel label “b” is used twice. One of them should be “c”

We have corrected it.

Figure 5: Panel label “c” is used twice. One of them should be “d”

We have corrected it.

Figure 5a: In the legend, the scale bar for expression is lacking a gradient.

We have corrected it.

Supp. Fig. 5b: In the legend, the scale bar for % match is lacking a gradient.

We have corrected it.

Fig. 7: label for panel g is missing.

Fig.7 has been updated.

Fig. 7 f: Do the blue dots indicated common master regulators shared with the lamprey? If so, please indicate this in the figure legend.

Yes. We have indicated this in the updated figure legend (**Supplementary Fig. 11**).

Line 1229: (e) should be (c)

Due to the revisions in Figure 7, the legend has been updated accordingly.

General notes:

Please italicise all gene names in the text and figures.

We have italicized all gene names in the text and figures.

Methods:

I see all scripts will be available on GitHub. To allow analysis to be easily repeated, please refer to specific script file names at the appropriate location in the methods section so it is clear which script was used for each part of your analysis.

Thank you for the good suggestion. In the GitHub folder, we have archived different analyses into distinct subfolders and provided detailed README documents as usage introduction.

Line 880: Please provide primer and probe template sequences

We have provided primers and probe template sequences in **Supplementary Table 3**.

Line 878: Please add the reference number for Peng 2019.

We have added it.

Line 883: How long was the Proteinase K treatment done for?

5 mins. We have added this information in the **Methods “Fluorescence in Situ Hybridization (FISH) Validations”** section.

Line 884-885: Was Peng 2019 also followed for the in situ hybridisation procedure? If so, please add the reference here too or if not, give the appropriate reference.

We have added the reference.

Line 906: how was the RNA obtained? Please provide tissue dissociation and RNA extraction details.

We have provided these details.

Line 914: spicing should be splicing

We have corrected it.

Line 924: Please provide details and scripts explaining how this annotation was done

We have added additional details to explain how the annotation was performed.

Line 932: Please provide details of the retina dissociation protocol and details of the version of the 10x Genomics kit used. How many cells were targeted and how many cycles were used during the amplification steps?

We have provided these details in the **Methods “Single-Cell RNA-sequencing Library Preparation.”**

Line 950 and line 979: What were the parameters for removing low quality cells?

We have provided the parameters for removing low quality cells in the **Methods**.

Line 983: Please provide the details of the mouse and chicken datasets used, including links to the appropriate data repositories.

We have provided the details including the links to the published datasets we used in this study in **Supplementary Table 2**.

Line 996 and line 1076: What were the downsampling parameters?

We have provided the parameters in **Supplementary Table 6, and 7**.

RESPONSE TO REVIEWERS' COMMENTS

Reviewer #2 (Remarks to the Author):

Dear Editor,

the authors have very well addressed all my concerns. Thank you! I am satisfied with the revisions and think it is now ready for publishing.

Thank you! We greatly appreciate all the suggestions and comments you provided during the review of our study. Your input has significantly contributed to improving the quality of this manuscript.

Reviewer #3 (Remarks to the Author):

The responses to my comments, new analyses and updates to the manuscript that the authors have provided have dealt with the majority of my concerns. I am grateful for their extra work.

I have some minor concerns about how they have addressed some of my comments and a few new, minor suggestions for corrections. I have copied my original comments and their responses below. I have added my new comments related to each point in **bold**.

Thank you for the positive assessment and valuable suggestions. We have made all the corrections.

Line 107: How many genes showed an improvement/change in their gene-body definitions? After the update, the NCBI+TruSeq reference transcriptome had 15,070 genes updated by StringTie with MSTRG numbers, suggesting changes in their gene body definitions (Supplementary Fig. 1c). A detailed comparison between the NCBI and NCBI+TruSeq references using gffcompare revealed 15,871 novel exons, 11,391 novel introns, and 3,430 novel loci in the NCBI+TruSeq reference transcriptome.

Thank you for this additional gffcompare analysis which addresses my comment. Please include this information in the manuscript. Also, please note, the reference to Supplementary Fig. 1c is no longer in the manuscript. Please add this back.

We have included this information in the Results section and also referenced Supplementary Fig. 1c.

Line 117: How many cells did you obtain per replicate?
We obtained 12,021 and 9,453 cells from the S1 and S2 replicates, respectively.

Please add this information to line 115.

We have added this information.

Line 176-178: It would be useful to use your analysis of the mouse scRNA-seq data to show that the equivalent cell type in mice does not express SLC17A6 to support this statement.

The lack of SLC17A6 expression in BCs, while present in RGCs, has been demonstrated in

Figure 5D of Macosko et al., 2015 (Ref 29) and Figure 1C of Yan et al., 2020 (Ref 34). We have included these references in the corresponding statement.

Please add ref 29 to Line 185. It is missing.

We have added ref 29 accordingly.

Line 247-250 and Fig. 5a: SLC6A5 is not expressed across all the AC groups, suggesting there may be distinct subgroups. Your reference 58 (Yan et al 2020) suggests that Gad1 and Gad2 expression profiles most strongly discriminate between AC subgroups. What do these markers look like in your data? This analysis is needed to support your conclusions.

We apologize for the confusion. We have revised the sentences. GAD2 is not found in the lamprey genome. We have plotted the expression of GAD1, SLC32A1 (VGAT), SLC6A1 (GAT1), SLC6A11 (GAT3), SLC6A9 (GLYT1), and SLC6A5 (GLYT2) in all AC types in Supplementary Fig. 6b. SLC6A13 (GAT2) is not found in the lamprey genome, and the expression of SLC6A1 is very low in all AC types.

All AC types except AC8 express both SLC6A11 (GAT3) and SLC6A9 (GLYT1), suggesting they are both GABAergic and glycinergic. AC8 expresses only SLC6A5, indicating that it is a glycinergic AC type.

In line 279 in the revised manuscript you state: “AC8 is uniquely glycinergic with the expression of only the glycine transporter SLC6A9 (GLYT1)”. Given your statement in response to my comment above and the data in Supplementary Fig. 6, which shows SLC6A9 is not expressed by AC8, this should be SLC6A5 (GLYT2). Please correct this.

We have corrected the error.

Interestingly, among all the GABAergic and glycinergic ACs, AC1, AC6, AC7, AC9, AC10, and AC11 express both glycine transporter SLC6A5 and SLC6A9, while AC2-5 only express one glycine transporter SLC6A9.

In line 280, AC7 is missing from this list. Please add it in.

We have added AC7 to the list.

Line 288: This statement contradicts your statement in Line 273-274 about ipRGCs. Also, it is not clear how you came to this conclusion. There are no ipRGCs listed in the comparison analysis in Fig. 6b and you have not shown a comparison between lamprey and chicken RGCs. This conclusion needs to be backed up with stronger evidence.

We have showed that RGC5 and 13 have the highest correspondence percentage with mouse M1 ipRGCs. We also included the comparison of RGCs between lamprey and chicken and also between lamprey and zebrafish.

Thank you for this clarification. It would be helpful to add the specific names of the M1 ipRGCs that are used in the figures to line 327 so it is clear which cell types you are referring to.

We have added the specific names of mouse M1 ipRGC types to the Results section.

New comments:

• **Line 297-298:** The violin plots in Supp. Fig. 6c indicate that some cells in these groups do express these genes. The sentence should rather read: “We detected very little expression of any other lamprey gap-junctional proteins in AC8”

We have corrected the sentence.

• **Supplementary Figure 6 legend:** Panels “b” and “c” show violin plots, not dot plots. Please correct.

We have corrected the errors.

• **You have not referred to Supplementary figure 9 in the text. Please add it in.**

We have referenced Supplementary Fig. 9 in the Results section.

• **Line 1280:** Should this be Supplementary Table 7? There is no Supplementary Table 9.

We have corrected the error; it is now Supplementary Table 5.

Reviewer #3 (Remarks on code availability):

The link is not working.

The link is now active and public.